# INSTRUCTING LARGE LANGUAGE MODELS TO IDENTIFY AND IGNORE IRRELEVANT CONDITIONS

## ABSTRACT

Math word problem (MWP) solving requires generating a reasoning path based on a given problem description that often contains *irrelevant conditions*. Existing chain-of-thought (CoT) prompting methods elicited multi-step reasoning abilities of large language models (LLMs) to solve MWPs. However, they were seriously confused by the irrelevant conditions, resulting in low accuracy. In this paper, we propose a novel approach named $I^3C$ that instructs LLMs to identify and ignore irrelevant conditions. It identifies a set of irrelevant condition candidates that have a weak semantic relevance with the question. Then it prompts LLMs to verify the irrelevant conditions. Lastly it instructs the LLMs with the verification on relevant and irrelevant conditions to avoid confusion and improve reasoning paths. Moreover, we propose to select (problem, reasoning paths)-pairs as demonstrations to enhance $I^3C$ with few-shot reasoning. We develop $I^3C$-Select that selects the most confusing problems based on the semantic relevance measurement. We conduct extensive experiments on six MWP datasets. $I^3C$ can be combined with any CoT prompting methods to improve the performance of solving MWPs. Notably, $I^3C$-Select achieves an accuracy of 93.7 and 90.9 on GSM-IC2-1K and GSM-ICM-1K, respectively, significantly outperforming the state-of-the-art few-shot prompting method Auto-CoT by $+19.4$ and $+25.7$.

## 1 INTRODUCTION

Math word problem (MWP) solving is a task of developing algorithms to generate a reasoning path towards an unknown quantity based on a problem description. This task is challenging as it requires mathematical understanding and multi-step reasoning abilities. Chain-of-thought (CoT) prompting methods were able to guide large language models (LLMs) to perform complex multi-step reasoning (Kojima et al., 2022; Wang et al., 2023a). Adding demonstrations created manually Wei et al. (2022) or retrieved from a large training set Zhang et al. (2023) in CoT prompts enabled few-shot in-context learning and improved accuracy. However, Shi et al. found that LLMs could be seriously confused by irrelevant conditions which are specifications or data presented in a problem that are unrelated to the solution (Kellogg, 2016). For example, as shown in Figure 1a, the third condition "*The height of Mary is 5 feet.*" was irrelevant to the final question and misled the reasoning and prediction. Shi et al. added a plain instruction "*Feel free to ignore irrelevant conditions in the problem description.*" in the prompts, but the LLMs could not effectively ignore them in the problem solving process because they were not identified or specified in the instruction.

Improving the reasoning on MWPs that have irrelevant conditions is non-trivial. Self-consistency (Wang et al., 2023b) was proposed to repeatedly solve a problem multiple times (e.g., 10 times) and employ a majority vote strategy to determine the most consistent answer as the final answer. However, it was computationally expensive and still confused by the irrelevant conditions. Moreover, the demonstrations would have to be re-designed to obtain the few-shot learning ability of identifying and ignoring the irrelevance, compared to those in (Wei et al., 2022; Zhang et al., 2023).

In this paper, we propose a novel approach named $I^3C$ to instruct LLMs to explicitly Identify and Ignore Irrelevant Conditions in the mathematical reasoning process. It creates effective instructions that can be added to any CoT prompts to improve their generated reasoning paths. Unlike self-consistency, $I^3C$ does not prompt LLMs many times. Its advanced variant $I^3C$-Select uses the most confusing problems and their generated reasoning paths as demonstrations for few-shot learning.

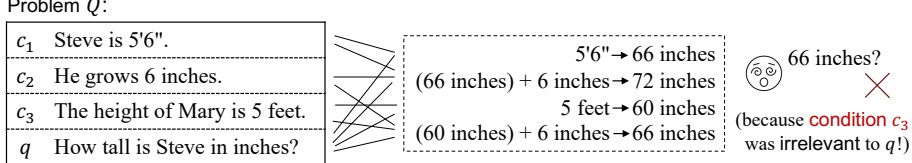

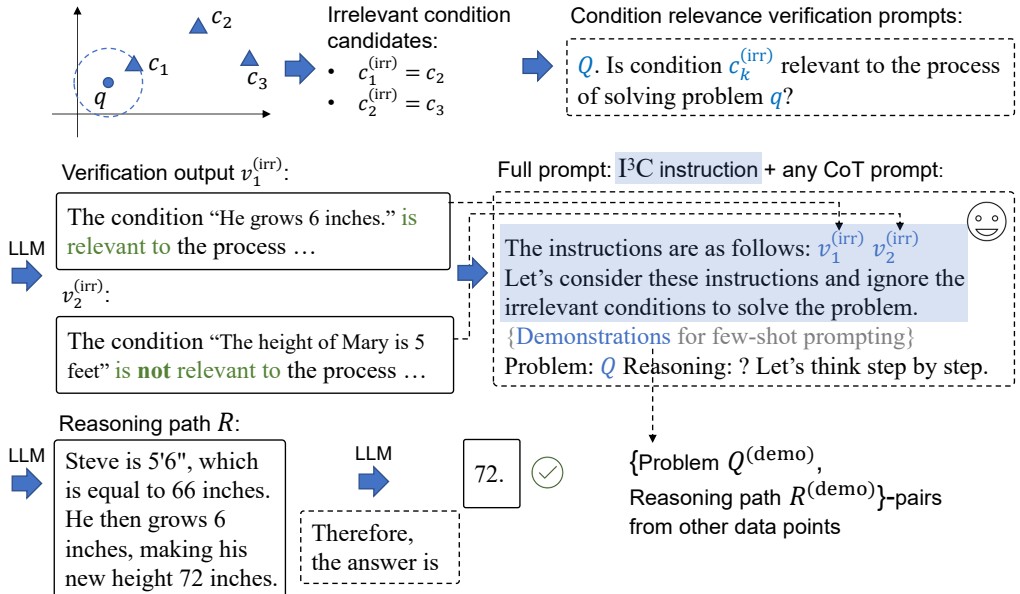

(a) LLMs were confused by irrelevant conditions in complex math word problems and gave wrong answers.

(b) I³C performs three steps: (1) Identify irrelevant condition candidates by encoding and condition-question similarity scoring; (2) Use LLMs to verify if the candidates are relevant; (3) Leverage the verifications (and demonstrations) to generate accurate reasoning paths and find correct answers.

Figure 1: The proposed I³C approach instructs LLMs to Identify and Ignore Irrelevant Conditions.

First, we quantify the semantic relevance of each condition $c_i$ in a MWP $Q = [\{c_i\}, q]$. Specifically, we use a pre-trained language model like SimCSE (Gao et al., 2021) to encode the conditions $\{c_i\}$ and question sentence $q$. The semantic relevance is lower if the condition's encoding is more distant from the encodings of question and other conditions, as shown in Figure 1b. Then we identify a set of irrelevant condition candidates, like $c_2$ and $c_3$ in this example, and we denote them by $\{c_k^{(irr)}\}$.

Next we use a LLM to verify if the candidates are indeed irrelevant. For each candidate $c_k^{(irr)}$, the verification prompt is a natural language question consisted of itself, $Q$, and $q$. The verification output usually has the explicit answers "... *is (not) relevant to ...*", denoted by $v_k^{(irr)}$.

Finally we put all the verification outputs $\{v_k^{(irr)}\}$ to create a novel instruction which helps the LLM to identify and ignore irrelevant conditions in the problem description, so-called I³C. The I³C instruction is a plug-and-play module and can be added to any CoT prompting methods to help LLMs avoid confusion and improve their generated reasoning paths.

To enable few-shot in-context learning, we further develop I³C-Select that uses the pairs of solved problems and generated reasoning paths to automatically construct effective demonstrations. Specifically, it defines the confusion score of each solved problem: the score is higher, if the semantic relevance of its conditions is lower; and the problems of the highest confusion scores are selected.

Experiments on GPT-3 demonstrate that adding the I³C instruction to CoT prompting methods improves their performance. For example, adding I³C instruction to Manual-CoT improves the accuracy by +5.1 on AddSub, +3.4 on SVAMP, +4.7 on GSM8K, +2.6 on SingleEq, +8.1 on GSM-IC2-1K, and +5.5 on GSM-ICM-1K. Moreover, I³C-Select beats existing prompting methods by a striking margin on six MWP datasets. Specifically, I³C-Select boosts the performance of Zero-Shot-CoT method by +23.0 on GSM-IC2-1K, +28.4 on GSM-ICM-1K, and +12.0 on GSM8K.

## 2 RELATED WORK

### 2.1 MATH WORD PROBLEM SOLVING

Our work is related to existing efforts on solving MWPs. Traditional methods used statistical learning to extract entities, quantities, and operators from a question and generated an arithmetic equation to find the answer (Hosseini et al., 2014; Roy et al., 2015; Zhou et al., 2015; Mitra & Baral, 2016). Later, sequence-to-sequence (Seq2Seq) model and recurrent neural networks directly transformed the question into an arithmetic equation (Wang et al., 2017; 2019). Recently, fine-tuned pre-trained language models have significantly improved the validity of generated equations and accuracy of answers (Shen et al., 2021; Liang et al., 2022; 2023). However, these methods require a large amount of human annotations, lacking the ability to generalize to new kinds of MWPs. In this work, we aim to prompt LLMs to answer arbitrary MWPs without human annotations or task-specific fine-tuning. Our approach generates reasoning paths so that researchers can investigate the behaviors of LLMs.

### 2.2 CHAIN-OF-THOUGHT PROMPTING METHODS

CoT prompting methods have enabled LLMs to generate reasoning paths and solve complex MWPs (Kojima et al., 2022). The reasoning paths could be more expressive if the prompts were added with "*Let's think step by step*". To mitigate missing-step errors, Plan-and-Solve (PS) prompting methods instructed the LLMs to devise a plan to break down the entire task into smaller subtasks, and then carry out the subtasks according to the plan (Wang et al., 2023a). Manual-CoT, as a type of few-shot prompting, manually designed demonstrations to elicit multi-step reasoning ability of the LLMs (Wei et al., 2022). Program of Thought (PoT) generated programming language statements and used a program interpreter to execute the generated program to get final answers (Chen et al., 2022a). Zhang et al. (2023) designed Auto-CoT that sampled diverse questions from data and minimized the manual effort on finding demonstrations. Aware of irrelevant conditions in the problem description, Instruct-CoT added the instruction "*Feel free to ignore irrelevant conditions in the problem description*" in the prompt (Shi et al., 2023). These methods do not explicitly specify the irrelevant conditions in the prompt, which makes it difficult for LLMs to identify and ignore irrelevant conditions in the problem solving process. Our method identifies irrelevant conditions in the problem description, instructs the LLMs to ignore them, and achieves significantly higher accuracy.

### 2.3 IDENTIFY AND IGNORE IRRELEVANT INFORMATION

Jia & Liang (2017) have shown that neural question answering systems are confused when paragraphs contain irrelevant information. Several studies have trained models to identify and filter out the irrelevant information. For example, Roy & Roth (2015) trained a classifier and scored the likelihood of each quantity in the problem being an irrelevant quantity. Kim et al. (2022) employed a new training loss to remove the attribute-irrelevant information from the semantic encoder output. Li et al. (2022) proposed a multi-scale knowledge-aware transformer to eliminate identity-irrelevant information. Yang et al. (2023) leveraged pre-extracted semantic information to improve the preprocessor's ability to accurately identify and filter out task-irrelevant information. All these methods require massive human annotations. In contrast, our method does not require time-consuming training or fine-tuning. It employs large language models to automatically identify irrelevant conditions and generate instructions to help the models ignore them.

## 3 PROPOSED APPROACH

### 3.1 OVERVIEW

In this section, we elaborate on how to instruct LLMs to identify and ignore irrelevant conditions in the math word problem description. Given a complex problem, we first identify a set of irrelevant condition candidates that have a weak semantic relevance with the question (§ 3.2). Then we prompt LLMs to verify if the candidates are indeed irrelevant. Putting all the verification results together, we create a novel $I^3C$ instruction to instruct the LLMs to ignore the irrelevant conditions in the problem description. The $I^3C$ instruction can be added to any CoT prompting methods to help LLMs avoid confusion and improve their generated reasoning paths. Furthermore, we develop a few-shot

115 prompting method I³C-Select that selects the most confusing problems and their reasoning paths as
116 demonstrations, and adds the I³C instruction before the demonstrations in the prompt. Given the
117 prompt and a target problem, the LLMs generate an accurate reasoning path to improve the solving
118 process. We introduce the I³C instruction in § 3.3 and I³C-Select method in § 3.5.

## 3.2 IDENTIFY A SET OF IRRELEVANT CONDITION CANDIDATES

120 Given a math word problem $Q$, we first split it into $n$ conditions $\{c_i\}_{i=1}^n$ and a question sentence
121 $q$, where each condition describes at most one quantity. So we have $Q = [\{c_i\}, q]$. For example,
122 in Figure 1a, the conditions are {"*Steve is 5'6".*", "*He grows 6 inches.*", "*The height of Mary is 30*
123 *feet.*"}, and the question sentence is "*How tall is Steve in inches?*".

124 Next, we use a pre-trained language model, e.g., SimCSE (Gao et al., 2021), to encode the con-
125 ditions and question sentence into vector representations. So we have $\{\mathbf{c}_i\}_{i=1}^n$ and $\mathbf{q}$ which are
126 $d$-dimensional vectors. We set $d = 1,024$.

127 Then for each condition $c_i$, we calculate the average similarity between $c_i$ and all other conditions
128 in $Q$ using cosine similarity, because the SimCSE embeddings were trained on cosine similarity:

$$s_i^{(\mathrm{c})} = \frac{1}{n-1} \sum_{j=1, j \neq i}^n \cos\left(\mathbf{c}_i, \mathbf{c}_j\right) = \frac{1}{n-1} \sum_{j=1, j \neq i}^n \frac{\mathbf{c}_i^\top \mathbf{c}_j}{\|\mathbf{c}_i\| \cdot \|\mathbf{c}_j\|}. \tag{1}$$

129 We also calculate the similarity between $c_i$ and $q$: $s_i^{(\mathrm{q})} = \cos\left(\mathbf{c}_i, \mathbf{q}\right)$. So we have $\{s_i^{(\mathrm{c})}, s_i^{(\mathrm{q})}\}_{i=1}^n$.

130 Now we can define a set of *irrelevant condition candidates* $\mathcal{I} \subset \{c_i\}_{i=1}^n$ for each math word prob-
131 lem. A condition $c_i$ is potentially irrelevant if its semantic relevance is lower than expectation. In
132 other words, if $s_i^{(\mathrm{c})} < \theta$ or $s_i^{(\mathrm{q})} < \theta$, $\mathcal{I}$ has $c_i$. We re-index the conditions in the set: $\mathcal{I} = \{c_k^{(\mathrm{irr})}\}_{k=1}^{|\mathcal{I}|}$.
133 The threshold $\theta$ is a hyperparameter. We set $\theta = 0.5$.

134 We can further define the *confusion score* of a math word problem $Q$. We assume that the problem
135 is more confusing if its conditions are less relevant with the final question. So the confusion score
136 is defined as the inverse of the average similarity between any condition and the question:

$$\mathrm{conf}(Q) = \left[\frac{1}{n} \sum_{i=1}^n \cos\left(\mathbf{c}_i, \mathbf{q}\right)\right]^{-1}. \tag{2}$$

137 The most confusing problems, i.e., the problems of the highest confusion scores, and their generated
138 reasoning paths, will be automatically used as demonstrations in a few-shot setting. The demos
139 teach LLMs to better solve confusing problems. Later sections give details.

## 3.3 CONSTRUCT I³C INSTRUCTION

141 Given a set of irrelevant condition candidates $\mathcal{I}$, we use a LLM to verify if the candidates are indeed
142 irrelevant. For a math word problem $Q$, its final question $q$, and a condition candidate $c_k^{(\mathrm{irr})} \in \mathcal{I}$,
143 we construct a verification prompt: "*$Q$. Is condition $c_k^{(\mathrm{irr})}$ relevant to the process of solving problem*
144 *$q$?*" We feed the prompt to a LLM and receive a piece of text $v_k^{(\mathrm{irr})}$ justifying if $c_k^{(\mathrm{irr})}$ is relevant or
145 indeed irrelevant. So we have a set of verification outputs (size $|\mathcal{I}|$): $\{v_k^{(\mathrm{irr})}\}_{k=1}^{|\mathcal{I}|}$.

146 Now we can create a novel instruction to help LLMs identify and ignore irrelevant conditions in the
147 problem description. In a zero-shot setting, the instruction starts with all the verification outputs.
148 Specifically, this I³C instruction, simply denoted by $I$, is "*The instructions are as follows: $v_1^{(\mathrm{irr})} \cdots$*
149 *$v_{|\mathcal{I}|}^{(\mathrm{irr})}$. Let's consider these instructions and ignore the irrelevant conditions to solve the problem*".
150 In case where $\mathcal{I}$ is an empty set, we follow the Instruct-CoT method (Shi et al., 2023) and use the
151 sentence "*Feel free to ignore irrelevant conditions in the problem description*" as the instruction.

## 3.4 GENERATE REASONING PATHS AND ANSWERS WITH I³C INSTRUCTION

153 The I³C instruction can be added to any CoT prompting methods such as Zero-Shot-CoT (Kojima
154 et al., 2022), PS (Wang et al., 2023a), Instruct-CoT (Shi et al., 2023), Manual-CoT (Wei et al., 2022),

Table 1: Accuracy (%) comparison on six MWP datasets. I³C indicates that instructs LLMs to identify and ignore irrelevant conditions. Adding the I³C instruction to CoT prompting methods effectively improves performance. Selecting the most confusing problems and their generated reasoning paths as demonstrations for few-shot learning (i.e., I³C-Select) achieves state-of-the-art performance on all six MWP datasets. The best performance for each dataset is shown in bold.

| Method | Dataset | | | | | |
|---|---|---|---|---|---|---|
| (text-davinci-003) | AddSub | SVAMP | GSM8K | SingleEq | GSM-IC2-1K | GSM-ICM-1K |
| Direct | 89.3 | 65.2 | 15.0 | 84.6 | 22.8 | 9.0 |
| Direct + I³C | 92.4 (+3.1) | 74.5 (+9.3) | 49.7 (+34.7) | 92.7 (+8.1) | 82.6 (+59.8) | 66.9 (+57.9) |
| Zero-Shot-CoT | 84.8 | 74.3 | 60.8 | 89.5 | 70.7 | 62.5 |
| Zero-Shot-CoT + I³C | 91.7 (+6.9) | 75.9 (+1.6) | 61.3 (+0.5) | 93.7 (+4.2) | 84.7 (+14.0) | 71.4 (+8.9) |
| PS | 88.1 | 72.0 | 58.2 | 89.2 | 70.9 | 63.5 |
| PS + I³C | 91.4 (+3.3) | 75.6 (+3.6) | 61.1 (+2.9) | 93.1 (+3.9) | 84.8 (+13.9) | 69.4 (+5.9) |
| Instruct-CoT | 90.4 | 76.3 | 57.8 | 91.1 | 82.4 | 64.3 |
| Instruct-CoT + I³C | 91.8 (+1.4) | 77.0 (+0.7) | 61.0 (+3.2) | 92.7 (+1.6) | 84.7 (+2.3) | 71.3 (+7.0) |
| Manual-CoT | 87.8 | 76.7 | 56.9 | 91.3 | 73.9 | 60.6 |
| Manual-CoT + I³C | 92.9 (+5.1) | 80.1 (+3.4) | 61.6 (+4.7) | 93.9 (+2.6) | 82.0 (+8.1) | 66.1 (+5.5) |
| Auto-CoT | 90.6 | 77.8 | 58.9 | 90.9 | 74.3 | 65.2 |
| Auto-CoT + I³C | 93.7 (+3.1) | 80.0 (+2.2) | 61.9 (+3.0) | 93.5 (+2.6) | 83.9 (+9.6) | 68.2 (+3.0) |
| I³C-Select (Ours) | **96.0** | **80.9** | **72.8** | **94.3** | **93.7** | **90.9** |

and Auto-CoT (Zhang et al., 2023). The goal is to generate a reasoning path and answer a math word problem $Q$. For example, in Zero-Shot-CoT (Kojima et al., 2022), the prompt was "*Q: q. A: Let's think step by step.*" where $q$ is the final question of $Q$. By adding the I³C instruction to the Zero-Shot-CoT method, denoted by Zero-Shot-CoT+I³C in our experiments, the prompt becomes "*I. Q: q. A: Let's think step by step*". The full prompts in experiments can be found in Appendix A.4.

Finally, after the reasoning path is generated, we use prompt "*Therefore, the answer is*" to get the quantity prediction as the final answer for evaluation.

### 3.5 I³C-Select: Select Confusing Problems as Automatic Demonstrations

Fu et al. (2023) indicated that prompts with higher reasoning complexity achieve better performance on multi-step reasoning tasks. To further enhance the ability of LLMs to address the irrelevance of conditions, we develop a novel few-shot prompting method I³C-Select. As presented in § 3.2, it first calculates the confusion score of solved problems, defined in Eq.(2), and selects the $K$ most confusing problems ($K = 8$ in our experiments). Next, it uses the most confusing problems and their reasoning paths as demonstrations, denoted by $\{Q_1^{(demo)}, R_1^{(demo)}; \cdots ; Q_K^{(demo)}, R_K^{(demo)}\}$.

I³C-Select puts the demonstrations after the I³C instruction to construct the full prompt. Specifically, the prompt is "*I. Q: $Q_1^{(demo)}$ A: $R_1^{(demo)}$ $\cdots$ Q: $Q_K^{(demo)}$ A: $R_K^{(demo)}$ Q: Q. A:*". With the prompt and the target problem $Q$, the LLMs generate a reasoning path for $Q$. Figure 1b illustrates the details.

## 4 Experiments

### 4.1 Experimental Setup

**Datasets.** We use six math word problem (MWP) datasets for evaluation. AddSub (Hosseini et al., 2014), SingleEq (Koncel-Kedziorski et al., 2015), SVAMP (Patel et al., 2021), and GSM8K (Cobbe et al., 2021) are classical MWP datasets in which part of the problem description contains irrelevant conditions. GSM-IC2-1K (Shi et al., 2023) and GSM-ICM-1K (Shi et al., 2023) are challenging datasets that require multi-step reasoning, and each problem description contains irrelevant conditions. More detailed dataset information can be found in Appendix A.1.

**Baselines.** We compare our proposed I³C-Select prompting method with two types of prompting baselines: (1) Zero-shot baselines. We include Zero-Shot-CoT (Kojima et al., 2022), PS (Wang et al., 2023a), Instruct-CoT (Shi et al., 2023), and Direct (Kojima et al., 2022). The Direct baseline uses

Table 2: Accuracy (%) on GSM-IC-2K dataset, broken down by the number of reasoning steps required in the standard answer. The GSM-IC-2K dataset is formed by merging the GSM-IC2-1K dataset and the GSM-ICM-1K dataset. The best performance for each dataset is shown in bold.

| Method (text-davinci-003) | Accuracy by Steps (GSM-IC-2K) | | | | |
|---|---|---|---|---|---|
| | 2 Steps | 3 Steps | 4 Steps | $\geq$ 5 Steps | All |
| Direct | 22.8 | 15.3 | 5.7 | 4.9 | 15.9 |
| Direct + I$^3$C | 82.6 (+59.8) | 74.3 (+59.0) | 66.8 (+61.1) | 59.0 (+54.1) | 74.8 (+58.9) |
| Zero-Shot-CoT | 70.7 | 67.8 | 62.9 | 56.4 | 66.6 |
| Zero-Shot-CoT + I$^3$C | 84.7 (+14.0) | 75.9 (+8.1) | 73.9 (+11.0) | 64.5 (+8.1) | 78.1 (+11.5) |
| PS | 70.9 | 69.4 | 63.3 | 57.3 | 67.2 |
| PS + I$^3$C | 84.8 (+13.9) | 73.7 (+4.3) | 71.7 (+8.4) | 62.8 (+5.5) | 77.1 (+9.9) |
| Instruct-CoT | 82.4 | 68.1 | 65.7 | 59.0 | 73.4 |
| Instruct-CoT + I$^3$C | 84.7 (+2.3) | 78.3 (+10.2) | 69.3 (+3.6) | 65.4 (+6.4) | 78.0 (+4.6) |
| Manual-CoT | 73.9 | 68.1 | 52.3 | 59.3 | 67.3 |
| Manual-CoT + I$^3$C | 82.0 (+8.1) | 72.1 (+4.0) | 64.3 (+12.0) | 61.1 (+1.8) | 74.1 (+6.8) |
| Auto-CoT | 74.3 | 80.4 | 53.7 | 58.1 | 69.8 |
| Auto-CoT + I$^3$C | 83.9 (+9.6) | 73.2 (−7.2) | 68.6 (+14.9) | 62.5 (+4.4) | 76.1 (+6.3) |
| I$^3$C-Select (Ours) | **93.7** | **93.3** | **90.1** | **89.0** | **92.3** |

the prompt "*The answer is*" to get the final answer. (2) Few-shot baselines. We include Manual-CoT (Wei et al., 2022) and Auto-CoT (Zhang et al., 2023). The demonstrations of the few-shot baselines are from their original papers. Details of all baselines are shown in Appendix A.2.

**Implementation.** We use GPT-3 (text-davinci-003) as the backend LLM, which is one of the most widely-used LLMs with public APIs[1]. Following (Shi et al., 2023), we set the temperature to 0.7 throughout our experiments. To evaluate the model performance, we follow (Chen et al., 2022b) to adopt accuracy as our evaluation metric. An answer is considered correct if and only if the absolute error between the answer and the gold answer is less than $1 \times 10^{-5}$. See Appendix A.3 for detail.

## 4.2 EXPERIMENTAL RESULTS

**Overall performance on MWP datasets.** As shown in Table 1, I$^3$C-Select consistently outperforms the baseline methods across all MWP datasets by a significant margin. Specifically, it improves the accuracy over Zero-Shot-CoT by at least +6.6 for all datasets, except for SingleEq, which has a +4.8 improvement. This exception can be attributed to the fact that the problems in SingleEq do not contain irrelevant conditions, and our proposed I$^3$C-Select method mainly instructs LLMs to identify and ignore irrelevant conditions in the problem description. It is worth noting that even in the SingleEq dataset, using the most confusing problems and their reasoning paths as demonstrations effectively enhances MWP solving performance.

In comparison to the competitive zero-shot baseline, Instruct-CoT, the performance of I$^3$C-Select remains impressive. It enhances the average accuracy by +11.0 across six MWP datasets, surpassing the Instruct-CoT prompting method. Furthermore, our analysis demonstrates that I$^3$C-Select outperforms few-shot baselines on all datasets. Specifically, when compared to the Auto-CoT prompting method, I$^3$C-Select exhibits superior performance in GSM-ICM-1K, GSM-IC2-1K, and GSM8K, with improvements of +25.7, +19.4, and +13.9, respectively. These findings indicate that incorporating more detailed instructions (e.g., I$^3$C instruction) and the most confusing problems and their reasoning paths into the prompt can achieve better performance.

**Does adding the I$^3$C instruction work?** As shown in Table 1, adding the I$^3$C instruction to the CoT prompting methods significantly improves the MWP solving performance. Specifically, adding the I$^3$C instruction to the Zero-Shot-CoT method (i.e., Zero-Shot-CoT+I$^3$C) improves the average accuracy by +6.0 on six MWP datasets, compared to the original Zero-Shot-CoT prompting method. For GSM-IC2-1K and GSM-ICM-1K, which contain irrelevant conditions in each problem description, Zero-Shot-CoT+I$^3$C improves the accuracy by +14.0 and +8.9, respectively. Even for

---

[1]Public API available at https://openai.com/api/.

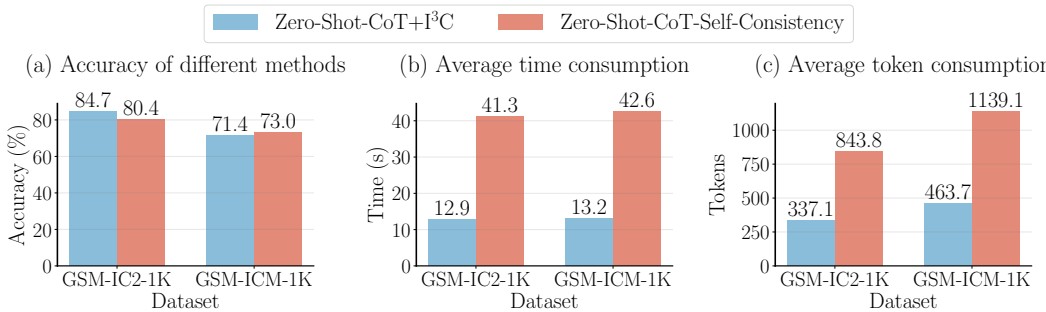

Figure 2: Performance comparison of Zero-Shot-CoT with I³C instruction (i.e., Zero-Shot-CoT+I³C) and Zero-Shot-CoT with self-consistency (i.e., Zero-Shot-CoT-Self-Consistency).

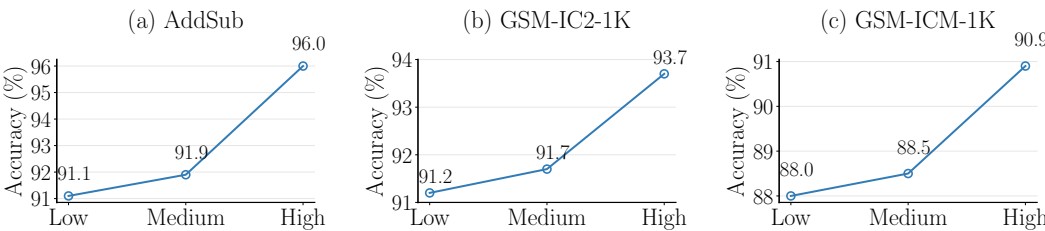

Figure 3: Comparison to other demonstration construction methods. "Low" indicates selecting eight problems with the lowest confusion scores. "Medium" indicates randomly selecting eight problems. "High" indicates selecting eight problems with the highest confusion scores.

prompting methods (e.g., Auto-CoT) that already achieve high accuracy on most MWP datasets, the addition of the I³C instruction (i.e., Auto-CoT+I³C) still leads to significant improvements. Auto-CoT+I³C improves accuracy by +9.6 on GSM-IC2-1K, and +3.0 on GSM8K.

**Does I³C instruction work for complex problems?** We analyze the breakdown accuracies for problems with respect to the reasoning steps[2] in Table 2. The GSM-IC-2K dataset is formed by merging the GSM-IC2-1K and GSM-ICM-1K datasets. Each problem in GSM-IC-2K contains irrelevant conditions and requires multiple steps to solve. Prompts with I³C instruction outperform baseline methods in solving problems that require at least 4 steps. Moreover, compared to Manual-CoT, I³C-Select significantly improves the performance on GSM-IC-2K: from 67.3 to 92.3. These results indicate that adding I³C instruction to the prompt can effectively solve complex problems.

**Efficiency and effectiveness of I³C instruction.** Self-consistency (Wang et al., 2023b) is the process of repeatedly solving a problem $M$ times and using a majority vote strategy to determine the most consistent answer as the final answer. We evaluate the performance of Zero-Shot-CoT with self-consistency (i.e., Zero-Shot-CoT-Self-Consistency) on the GSM-IC2-1K and GSM-ICM-1K datasets. Following (Wang et al., 2023a), we set $M$ to 10. As shown in Figure 2, adding the I³C instruction to Zero-Shot-CoT (i.e., Zero-Shot-CoT+I³C) consumes much fewer computational resources compared to Zero-Shot-CoT-Self-Consistency, while maintaining comparable accuracy.

### 4.3 ABLATION STUDIES

**How does demonstration construction affect I³C-Select?** In I³C-Select, we select the $K$ most confusing problems and their reasoning paths as demonstrations and named this demonstration construction method "High". To verify the effectiveness of the demonstration construction method, we also condiser: (1) "Low", where we select the $K$ problems with the lowest confusion scores and their reasoning paths as demonstrations, and (2) "Medium", where we randomly select $K$ problems and their reasoning paths as demonstrations. For a fair comparison, we set $K$ to 8 throughout our

---

[2]The number of reasoning steps of a problem is given by the number of sentences in its standard answer. (Cobbe et al., 2021)

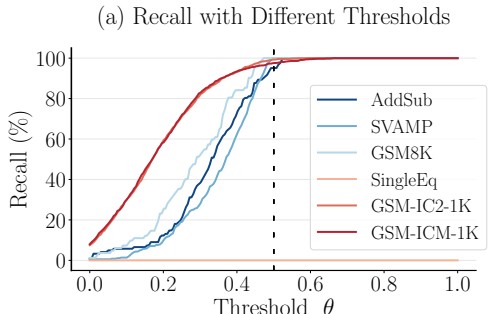 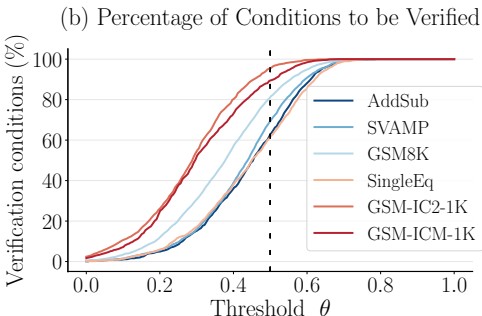

Figure 4: Hyperparameter analysis. (a) As the threshold increases, the recall scores of identified irrelevant condition candidates first increase and then remain unchanged for all datasets except SingleEq. (b) As the threshold increases, the percentage of conditions to be verified first increases and then remains unchanged for all datasets.

Table 3: Accuracy (%) when solving MWPs using I³C-Select with different LLMs.

| Method | LLM | AddSub | SingleEq | GSM-IC2-1K | GSM-ICM-1K |
|---|---|---|---|---|---|
| I³C-Select | text-davinci-002 | 80.0 | 84.8 | 87.2 | 87.3 |
| | text-davinci-003 | 96.0(+16.0) | 94.3(+9.5) | 93.7(+6.5) | 90.9(+3.6) |

experiments. As shown in Figure 3, selecting more confusing problems and their reasoning paths as demonstrations can effectively improve the model's performance.

**How does LLM selection affect I³C-Select?**  Table 3 shows that I³C-Select works better when the LLM is more powerful. Specifically, on the AddSub dataset, the text-davinci-003 model demonstrates a +16.0 increase in accuracy compared to the text-davinci-002 model. Similarly, on the GSM-IC2-1K dataset, using the text-davinci-003 model leads to a +6.5 improvement in accuracy over the text-davinci-002 model. Notably, the text-davinci-002 model is fine-tuned using supervised instruction tuning, while the text-davinci-003 is fine-tuned with reinforcement learning (Zheng et al., 2023). The improved performance with text-davinci-003 can be attributed to its enhanced power, making it better at understanding and employing the given prompt.

**Instructing to ignore irrelevant conditions vs. refining problems to eliminate irrelevant conditions.**  In Zero-Shot-CoT+I³C, we use I³C instruction to instruct LLMs to identify and ignore irrelevant conditions in the MWP solving process. In addition, we can refine the given problem to eliminate irrelevant conditions based on the verification outputs generated in § 3.3, and solve the refined problem using the Zero-Shot-CoT method (i.e., Zero-Shot-CoT+Refine). As shown in Table 4, Zero-Shot-CoT+Refine (87.6, 77.4, and 64.8) substantially outperforms Zero-Shot-CoT (84.8, 70.7, and 62.5) on AddSub, GSM-IC2-1K, and GSM-ICM-1K, respectively. This highlights that the generated verification outputs can explicitly identify irrelevant conditions in the problem description. Furthermore, Zero-Shot-CoT+I³C consistently outperforms Zero-Shot-CoT+Refine. This is mainly because the identified irrelevant conditions may contain some useful conditions. When we refine the given problem, we may eliminate some useful conditions, resulting in an incorrect answer. Instructing the LLM to ignore irrelevant conditions can effectively alleviate the problem of losing useful conditions during problem refinement. Case studies are provided in Appendix A.5.

**Hyperparameter setup for I³C instruction.**  To compare the identification accuracy of irrelevant conditions at different thresholds $\theta$, we employ the recall metric. Recall is computed by dividing the number of correctly identified irrelevant conditions by the total number of irrelevant conditions. As shown in Figure 4a, as the threshold increases, the recall score for the irrelevant condition identification first increases and then remains unchanged in all datasets except SingleEq. The exception is due to the problems in SingleEq not containing irrelevant conditions, the recall score remains 0 regardless of the threshold. Notably, when the threshold is 0.5, we can accurately identify 98.6% of irrelevant conditions. Furthermore, as shown in Figure 4b, the percentage of conditions to be verified initially increases and then stabilizes across all datasets as the threshold increases. The per-

Table 4: Accuracy (%) comparison of different methods that help LLMs ignore irrelevant conditions.

| Method | Dataset | | |
|---|---|---|---|
| (text-davinci-003) | AddSub | GSM-IC2-1K | GSM-ICM-1K |
| Zero-Shot-CoT | 84.8 | 70.7 | 62.5 |
| Zero-Shot-CoT + Refine | 87.6 | 77.4 | 64.8 |
| Zero-Shot-CoT + I$^3$C | **91.7** | **84.7** | **71.4** |

Table 5: Case study of answers generated by different methods. Irrelevant conditions in the problem description is highlighted with red color, and the LLM-generated final answer is highlighted with yellow color. The gold answer is indicated in bold.

| | |
|---|---|
| **Problem** | Granger went to the grocery store. He saw that the Spam is $3 per can, the peanut butter is $5 per jar, and the bread is $2 per loaf. Jack owns 5 bottles of water with each worth $2. If Granger bought 12 cans of spam, 3 jars of peanut butter, and 4 loaves of bread, how much is the total amount he paid? |
| **Gold Answer** | $3 \times 12 + 5 \times 3 + 2 \times 4 = \mathbf{59}$ |
| **Auto-CoT** | Granger bought 12 cans of spam. 12 * $3 = $36. Granger bought 3 jars of peanut butter. 3 * $5 = $15. Granger bought 4 loaves of bread. 4 * $2 = $8. Jack owns 5 bottles of water. 5 * $2 = $10. The total amount he paid is $36 + $15 + $8 + $10 = $69. The answer is 69. ✗ |
| **Instruct-CoT** | Granger bought 12 cans of Spam, so that's 12 x $3 = $36. He bought 3 jars of peanut butter, so that's 3 x $5 = $15. He bought 4 loaves of bread, so that's 4 x $2 = $8. Jack owned 5 bottles of water, so that's 5 x $2 = $10. Adding them all up, the total amount Granger paid is $36 + $15 + $8 + $10 = $69. ✗ |
| **I$^3$C-Select** | First, calculate the cost for each item that Granger bought: 12 cans of Spam = 12 x $3 = $36; 3 jars of peanut butter = 3 x $5 = $15; 4 loaves of bread = 4 x $2 = $8. Then, calculate the total cost of all the items: $36 + $15 + $8 = $59. ✓ |

centage of conditions to be verified is calculated by dividing the number of conditions in the set of irrelevant condition candidates by the total number of conditions. When the threshold is $0.5$, the LLM is required to verify $66.6\%$ of conditions on average. Overall, we set the threshold $\theta$ to $0.5$ to achieve a trade-off between efficiency and effectiveness.

### 4.4 CASE STUDY

We present a real case study from the GSM-ICM-1K dataset in Table 5. It is evident that, apart from I$^3$C-Select, the other methods fail to provide accurate answers to the given problem. Instruct-CoT and Auto-CoT produce incorrect answers due to the incorporation of irrelevant conditions in the MWP solving process. In contrast, I$^3$C-Select explicitly identifies and ignores irrelevant conditions in the MWP solving process. Additional case studies can be found in Appendix A.6.

### 5 CONCLUSION

In this paper, we present a novel approach named I$^3$C to instruct LLMs to explicitly identify and ignore irrelevant conditions in the mathematical reasoning process. It first identifies a set of irrelevant condition candidates that have a weak semantic relevance with the question, and then prompts a LLM to generate verification outputs to verify if candidates are indeed irrelevant. By incorporating all the verification outputs, we obtained the I$^3$C instruction. The I$^3$C instruction is a plug-and-play module that can be added to any CoT prompting methods to help LLMs avoid confusion and improve their generated reasoning paths. Moreover, we present a novel few-shot prompting method, I$^3$C-Select, which selects the most confusing problems and their reasoning paths as demonstrations, and adds the I$^3$C instruction before the demonstrations to construct the prompt. Extensive experiments demonstrate that adding the I$^3$C instruction to CoT prompting methods effectively improves MWP solving performance, achieving new state-of-the-art performance on all MWP datasets.

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

# A APPENDIX

## A.1 DATASETS

We use six math word problem datasets for assessing prompting method quality. The statistics of the datasets are shown in Table 6. We give a brief description of the datasets used below:

− SingleEq (Koncel-Kedziorski et al., 2015) contains a set of grade-school algebra word problems. Every problem may involve multiple math operations including multiplication, division, subtraction, and addition.

− AddSub (Hosseini et al., 2014) consists of math word problems on addition and subtraction for third, fourth, and fifth graders.

− SVAMP (Patel et al., 2021) consists of one-unknown math word problems which can be solved by expressions requiring no more than two operators.

− GSM8K (Cobbe et al., 2021) consists of high quality grade school math word problems created by human problem writers. These problems take between 2 and 8 steps to solve, and solutions primarily involve performing a sequence of elementary calculations using basic arithmetic operations to reach the final answer.

− GSM-IC (Shi et al., 2023) is an arithmetic reasoning dataset with irrelevant conditions in the problem description. It is divided into two splits: GSM-IC2, consisting of problems requiring two steps to solve, and GSM-ICM, consisting of problems requiring more than two steps to solve. Being mindful of the experiment costs, we uniformly sample $1,000$ examples from the GSM-IC2 dataset (denoted by GSM-IC2-1K) and $1,000$ examples from the GSM-ICM dataset (denoted by GSM-ICM-1K) for evaluation and analysis purposes throughout this paper.

## A.2 BASELINES

As we study how to prompt large language models to solve math word problems, we employ seven prompting baselines. We give a brief description of the baselines used below:

− Direct (Kojima et al., 2022) is a baseline that utilizes the symbolic reasoning ability of large language models. By simply adding the sentence "*The answer is*" after the problem of interest, which instructs the large language model to generate the answer to the problem.

− Zero-Shot-CoT (Kojima et al., 2022) is a Chain-of-Thought prompting method. By adding "*Let's think step by step*" to the problem to elicit the large language model to generate reasoning path leading to the final answer.

− Plan-and-Solve (PS) (Wang et al., 2023a) replaces the sentence "*Let's think step by step*" with "*Let's first understand the problem and devise a plan to solve the problem. Then let's carry out the plan and solve the problem step by step*" to address the missing step issue in Zero-Shot-CoT.

− Instruct-CoT (Shi et al., 2023) adds the sentence "*Feel free to ignore irrelevant conditions in the problem description.*" before the problem of interest, which instructs the large language model to ignore irrelevant information in the problem description.

− Manual-CoT (Wei et al., 2022) is a few-shot prompting method. By representing manual designed demonstrations that solve the corresponding problems with intermediate reasoning steps in the prompts, Manual-CoT elicits multi-step reasoning ability of large language models.

− Auto-CoT (Zhang et al., 2023) automatically constructs demonstrations with questions and reasoning paths to eliminate manual designs in Manual-CoT.

## A.3 METRICS

We use accuracy to evaluate the performance of different prompting methods. Since large language models cannot perform the computation precisely (especially with high-precision floats), we consider an answer to be correct if and only if the absolute error between the answer and the gold answer

Table 6: Dataset description. The last column indicates the percentage of problems with irrelevant conditions in the problem description.

| Dataset | Number of Problems | Average Words | Irrelevant Condition |
|---|---|---|---|
| SingleEq | 508 | 27.4 | 0.0% |
| AddSub | 395 | 31.5 | 30.9% |
| SVAMP | 1,000 | 31.8 | 36.7% |
| GSM8K | 1,319 | 46.9 | 6.2% |
| GSM-IC2-1K | 1,000 | 41.8 | 100.0% |
| GSM-ICM-1K | 1,000 | 61.4 | 100.0% |

Table 7: Accuracy (%) comparison on six MWP datasets. $I^3C$ indicates that instructs LLMs to identify and ignore irrelevant conditions. Adding the $I^3C$ instruction to CoT prompting methods effectively improves performance. Selecting the most confusing problems and their generated reasoning paths as demonstrations for few-shot learning (i.e., $I^3C$-Select) achieves state-of-the-art performance on all six MWP datasets. The best performance for each dataset is shown in bold.

| Method (UL2-20B) | Dataset | | | | | |
|---|---|---|---|---|---|---|
| | AddSub | SVAMP | GSM8K | SingleEq | GSM-IC2-1K | GSM-ICM-1K |
| Direct | 28.6 | 16.9 | 5.0 | 21.7 | 12.9 | 9.5 |
| Direct + $I^3C$ | 33.9(+5.3) | 27.8(+10.9) | 9.8(+4.8) | 32.7(+11.0) | 21.3(+8.4) | 13.2(+3.7) |
| Zero-Shot-CoT | 32.9 | 29.5 | 22.7 | 38.8 | 29.6 | 25.5 |
| Zero-Shot-CoT + $I^3C$ | 36.7(+3.8) | 30.5(+1.0) | 22.7(+0.0) | 40.0(+1.2) | 40.6(+11.0) | 27.6(+2.1) |
| PS | 30.0 | 26.7 | 21.2 | 36.6 | 27.4 | 24.9 |
| PS + $I^3C$ | 31.9(+1.9) | 28.4(+1.7) | 21.3(+0.1) | 40.0(+3.4) | 32.4(+5.0) | 26.0(+1.1) |
| Instruct-CoT | 34.7 | 31.2 | 23.5 | 40.0 | 33.8 | 26.4 |
| Instruct-CoT + $I^3C$ | 35.4(+0.7) | 31.5(+0.3) | 21.2(−2.3) | 41.1(+1.1) | 40.0(+6.2) | 28.6(+2.2) |
| Manual-CoT | 34.9 | 31.7 | 25.2 | 43.3 | 35.4 | 28.0 |
| Manual-CoT + $I^3C$ | 39.0(+4.1) | 28.1(−3.6) | 22.2(−3.0) | 42.9(−0.4) | 43.0(+7.6) | 28.5(+0.5) |
| Auto-CoT | 36.7 | 31.9 | 24.5 | 41.9 | 35.0 | 29.4 |
| Auto-CoT + $I^3C$ | 39.5(+2.8) | 28.7(−3.2) | 24.7(+0.2) | 43.6(+1.7) | 41.1(+6.1) | 30.1(+0.7) |
| $I^3C$-Select (Ours) | **39.7** | **34.6** | **27.5** | **44.1** | **46.0** | **35.9** |

is less than $1 \times 10^{-5}$. Let $\mathcal{P}$ be a set of problems, the accuracy of the prompting method is

$$\text{Accuracy} = \frac{1}{|\mathcal{P}|} \sum_{Q \in \mathcal{P}} \mathbb{1}\left(a^{(\text{final})}, a^{(\text{gold})}\right)$$

$$\mathbb{1}\left(a^{(\text{final})}, a^{(\text{gold})}\right) = \begin{cases} 1, & \text{if } \text{Abs}\left(a^{(\text{final})} - a^{(\text{gold})}\right) < 1 \times 10^{-5} \\ 0, & \text{if } \text{Abs}\left(a^{(\text{final})} - a^{(\text{gold})}\right) \geq 1 \times 10^{-5} \end{cases}$$

where $a^{(\text{gold})}$ is the gold answer to question $Q$, $a^{(\text{final})}$ is the model-generated answer to question $Q$, and $\text{Abs}(\cdot)$ is the absolute value function.

## A.4 FULL PROMPTS IN EXPERIMENTS

We list the prompts for all experiments in Table 8.

## A.5 ADDITIONAL EXPERIMENTAL RESULTS

**Does $I^3C$ instruction work with weaker LMs?** In all our experiments in § 4, we use GPT-3 (text-davinci-003) as the backend LLM, but can $I^3C$ instruction work with weaker LMs? We compare CoT prompting methods with adding the $I^3C$ instruction to CoT prompting methods when use the UL2-20B (Tay et al., 2023) as backend LM. Note that UL2-20B is a weaker LMs with 20 billion parameters, but GPT3 has 175 billion parameters. As shown in Table 7, even though the absolute accuracies of UL2-20B are lower, adding the $I^3C$ instruction to CoT prompting methods effectively improves MWP solving performance, and $I^3C$-Select achieves consistent performance improvements on MWP datasets. This shows that $I^3C$ instruction can work with weaker LMs.

Table 8: All prompts used in experiments. $Q$ represents the problem to be solved. $I$ represents the I³C instruction that instructs LLMs to identify and ignore irrelevant conditions in the problem description. The demonstrations of Manual-CoT is from its original paper (Wei et al., 2022).

| Method | Prompt |
|---|---|
| Direct | *Q: Q*
*A: The answer is* |
| Direct + I³C | *I*
*Q: Q*
*A: The answer is* |
| Zero-Shot-CoT | *Q: Q*
*A: Let's think step by step* |
| Zero-Shot-CoT + I³C | *I*
*Q: Q*
*A: Let's think step by step* |
| PS | *Q: Q*
*A: Let's first understand the problem and devise a plan to solve the problem.*
*Then, let's carry out the plan and solve the problem step by step* |
| PS + I³C | *I*
*Q: Q*
*A: Let's first understand the problem and devise a plan to solve the problem.*
*Then, let's carry out the plan and solve the problem step by step* |
| Instruct-CoT | *Feel free to ignore irrelevant conditions in the problem description.*
*Q: Q*
*A: Let's think step by step* |
| Instruct-CoT + I³C | *I*
*Feel free to ignore irrelevant conditions in the problem description.*
*Q: Q*
*A: Let's think step by step* |
| Manual-CoT | {hand-crafted demonstrations}
*Q: Q*
*A:* |
| Manual-CoT + I³C | *I*
{hand-crafted demonstrations}
*Q: Q*
*A:* |
| Auto-CoT | {automatically constructed demonstrations}
*Q: Q*
*A:* |
| Auto-CoT + I³C | *I*
{automatically constructed demonstrations}
*Q: Q*
*A:* |
| I³C-Select (Ours) | *I*
{the most confusing problems and their reasoning paths}
*Q: Q*
*A:* |

**Instructing to ignore irrelevant conditions vs. refining problems to eliminate irrelevant conditions.** In Zero-Shot-CoT+I³C, we use I³C instruction to instruct LLMs to identify and ignore irrelevant conditions in the MWP solving process. In addition, we can refine the given problem to eliminate irrelevant conditions based on the verification outputs generated in § 3.3, and solve the refined problem using the Zero-Shot-CoT method (i.e., Zero-Shot-CoT+Refine). For example, as shown in Table 9, the condition "*On Friday, he only works from 8am to 11am.*" and the condition "*Last week, Tom repaired 10 more umbrellas than the cobbler.*" are identified as the irrelevant conditions. By eliminating these identified irrelevant conditions, we get the refined problem "*A cobbler can mend 3 pairs of shoes in an hour. From Monday to Thursday, the cobbler works for 8 hours each day. How many pairs of shoes can the cobbler mend in a week?*". Obviously, in this case, we incorrectly identified the condition "*On Friday, he only works from 8am to 11am.*" as an irrelevant condition. Eliminating this condition would result in losing useful conditions in the problem refinement process, resulting in an incorrect answer. In contrast, instructing the LLM to ignore ir-

Table 9: Example outputs by Zero-Shot-CoT + Refine and Zero-Shot-CoT + I$^3$C for GSM-ICM-1K. Irrelevant conditions in the problem description is highlighted with red color.

| | |
|---|---|
| **Problem** | A cobbler can mend 3 pairs of shoes in an hour. From Monday to Thursday, the cobbler works for 8 hours each day, and on Friday, he only works from 8am to 11am. Last week, Tom repaired 10 more umbrellas than the cobbler. How many pairs of shoes can the cobbler mend in a week? |
| **Gold Answer** | $(4 \times 8 + 11 - 8) \times 3 = 105$ |
| **Verification Outputs** | ■ Yes, the condition A cobbler can mend 3 pairs of shoes in an hour is **relevant** to the calculation process of the problem How many pairs of shoes can the cobbler mend in a week? Without the condition, it would be impossible to calculate the number of pairs of shoes the cobbler can mend in a week.

■ The condition on Friday, he only works from 8am to 11am is **not relevant** to the calculation process.

■ The condition Last week, Tom repaired 10 more umbrellas than the cobbler. is **not relevant** to the calculation process of the problem. |
| **Refined Problem** | A cobbler can mend 3 pairs of shoes in an hour. From Monday to Thursday, the cobbler works for 8 hours each day. How many pairs of shoes can the cobbler mend in a week? |

| **Reasoning Path** | **Zero-Shot-CoT + Refine** | **Zero-Shot-CoT + I$^3$C** |
|---|---|---|
| | First, we need to calculate how many pairs of shoes the cobbler can mend in 4 days (Monday to Thursday). Since he works 8 hours a day, and can mend 3 pairs of shoes per hour, he can mend 8 x 3 = 24 pairs of shoes in 4 days. This gives us 24 pairs of shoes in a week. | Step 1: Calculate the number of hours the cobbler works in a week. From Monday to Thursday, the cobbler works for 8 hours each day. That's 8 x 4 = 32 hours. On Friday, he only works from 8am to 11am. That's 3 hours. So in a week, the cobbler works 32 + 3 = 35 hours. Step 2: Calculate the number of pairs of shoes the cobbler can mend in a week. The cobbler can mend 3 pairs of shoes in an hour. So in 35 hours, the cobbler can mend 35 x 3 = 105 pairs of shoes. Step 3: Determine if the condition is relevant to the calculation process. The fact that Tom repaired 10 more umbrellas than the cobbler does not affect the calculation process. Therefore, the cobbler can mend 105 pairs of shoes in a week. |
| **Final Answer** | 24 ✗ | 105 ✓ |

relevant conditions can effectively alleviate the problem of losing useful conditions during problem refinement, and can effectively enhance the MWP solving performance.

**I$^3$C Instruction Improves Robustness to Irrelevant Conditions.** We present case studies in Tables 10 to 15. It is evident that I$^3$C instructs LLMs to explicitly identify and ignore irrelevant conditions in the mathematical reasoning process. Adding I$^3$C instruction to the CoT prompting methods effectively helps LLMs avoid confusion and improve reasoning paths.

## A.6 SAMPLE PREDICTIONS ON MWP DATASETS

Table 10: Example outputs by Zero-Shot-CoT, Zero-Shot-CoT + I³C and I³C-Select. Irrelevant conditions in the problem description is highlighted with red color, and the LLM-generated final answer is highlighted with yellow color.

| **Problem** | Megan is an actress. She was the lead actress in 80% of her work. In total, Megan participated in 100 plays. Tom bought the tickets for 300 plays. How many times Megan was not the lead actress? |
|---|---|
| **Gold Answer** | $100 \times (1 - 80\%) = 20$ |
| **Verification Outputs** | ■ Megan participated in 100 plays. That means she was the lead actress in 80 of them. That leaves 20 plays where she was not the lead actress. Tom bought tickets to 300 plays. That means Megan was the lead actress in 240 of them, and she was not the lead actress in 60 of them. So, the answer is 60 times. The condition She was the lead actress in 80% of her work is relevant to the calculation process, as it tells us that 80 out of 100 plays she was the lead actress. 

 ■ Megan was the lead actress in 80% of her plays, so she was the lead actress in 80 plays. Since she participated in a total of 100 plays, that means she was not the lead actress in the remaining 20 plays. The condition that Tom bought tickets for 300 plays is irrelevant to the calculation process. |
| **Reasoning Path** | **Zero-Shot-CoT** 

 Megan was the lead actress in 80 plays. 100 - 80 = 20 Tom bought the tickets for 300 plays. Megan was not the lead actress in the remaining 200 plays. ✗ 

 **Zero-Shot-CoT + I³C** 

 Megan was the lead actress in 80% of her work, which means she was the lead actress in 80 plays. That leaves 20 plays in which she was not the lead actress. Tom bought tickets for 300 plays, but that information does not affect the calculation of how many times Megan was not the lead actress. Therefore, the answer is 20 times. ✓ 

 **I³C-Select** 

 Megan was the lead actress in 80% of her work, so she was the lead actress in 80 plays. That leaves 20 plays in which she was not the lead actress. Tom bought the tickets for 300 plays, but that information does not affect the calculation of how many times Megan was not the lead actress. Therefore, the answer is 20 times. ✓ |

Table 11: Example outputs by Direct, Direct + I$^3$C and I$^3$C-Select. Irrelevant conditions in the problem description is highlighted with red color, and the LLM-generated final answer is highlighted with yellow color.

| Problem | Sally eats 2 sandwiches on Saturday and 1 sandwich on Sunday. Jack bought 2 more pieces of bread than Sally yesterday. If each sandwich uses 2 pieces of bread, how many pieces of bread does Sally eat across Saturday and Sunday? |
|---|---|
| **Gold Answer** | $(2+1) \times 2 = 6$ |
| **Verification Outputs** | ■ Sally eats 2 sandwiches on Saturday, so she has eaten 4 pieces of bread. On Sunday, she eats 1 sandwich, so she has eaten 2 more pieces of bread. Jack bought 2 more pieces of bread than Sally, so Sally has eaten 6 pieces of bread across Saturday and Sunday. Therefore, the answer is 6 pieces of bread. The condition of 1 sandwich on Sunday is relevant to the calculation process, since it tells us how many pieces of bread Sally has eaten on Sunday. 

 ■ Sally ate 2 sandwiches on Saturday, so she used 4 pieces of bread on Saturday. On Sunday, Sally ate 1 sandwich, so she used 2 pieces of bread on Sunday. Therefore, Sally ate 6 pieces of bread across Saturday and Sunday. The condition Jack bought 2 more pieces of bread than Sally is not relevant to the calculation process, as it does not affect the number of pieces of bread Sally ate across Saturday and Sunday. |
| **Reasoning Path** | **Direct** 
 Sally eats a total of 4 pieces of bread across Saturday and Sunday. ✗ 

 **Direct + I$^3$C** 
 6 pieces of bread. Sally ate 2 sandwiches on Saturday, which used 4 pieces of bread, and 1 sandwich on Sunday, which used 2 pieces of bread, so she ate 6 pieces of bread across Saturday and Sunday. The condition that Jack bought 2 more pieces of bread than Sally is not relevant to the calculation process. ✓ 

 **I$^3$C-Select** 
 Sally ate 2 sandwiches on Saturday, so she used 4 pieces of bread on Saturday. On Sunday, Sally ate 1 sandwich, so she used 2 pieces of bread on Sunday. Therefore, Sally ate 6 pieces of bread across Saturday and Sunday. The condition Jack bought 2 more pieces of bread than Sally is not relevant to the calculation process, as it does not affect the number of pieces of bread Sally ate across Saturday and Sunday. ✓ |

Table 12: Example outputs by PS, PS + I$^3$C and I$^3$C-Select. Irrelevant conditions in the problem description is highlighted with red color, and the LLM-generated final answer is highlighted with yellow color.

| Problem | Holden's current master bedroom is 309 sq ft and his master bath is 150 sq ft. The basketball field of Tom is 200 sq ft larger than Holden's bathroom. If Holden wants to add a home office/personal gym divided space that is twice as large as his bedroom and bathroom, how much sq ft will this new room have? |
|---|---|
| **Gold Answer** | $(309 + 105) \times 2 = 918$ |
| **Verification Outputs** | ■ The master bedroom is 309 sq ft and the master bathroom is 150 sq ft. Therefore, the combined area of the bedroom and bathroom is 459 sq ft. If this space is to be doubled for the new home office/personal gym, then the new room will have 918 sq ft. The condition Holden's current master bedroom is 309 sq ft is relevant to the problem, as it is used to determine the combined area of the bedroom and bathroom (459 sq ft). This combined area is then doubled to get the total area of the new room (918 sq ft).

■ Holden's bedroom is 309 sq ft. Holden's bathroom is 150 sq ft. The total sq ft of Holden's bedroom and bathroom is 459 sq ft. We need to find a space that is twice as large as Holden's bedroom and bathroom combined. Therefore, the new room will be 2 x 459 sq ft = 918 sq ft. The basketball field of Tom is 200 sq ft larger than Holden's bathroom so it is not relevant to the calculation process of this problem. |
| **Reasoning Path** | **PS**

Plan: 1. Find the combined sq ft of the bedroom and bathroom. 2. Add 200 sq ft to the combined sq ft from step 1. 3. Multiply the combined sq ft from step 2 by 2. Solution: 1. 309 sq ft (bedroom) + 150 sq ft (bathroom) = 459 sq ft 2. 459 sq ft + 200 sq ft = 659 sq ft 3. 659 sq ft x 2 = 1318 sq ft Therefore, the new room will have 1318 sq ft. ✗

**PS + I$^3$C**

First, we need to calculate the combined area of Holden's bedroom and bathroom, which is equal to 309 sq ft (bedroom) + 150 sq ft (bathroom) = 459 sq ft. Next, we need to double this combined area to get the total area of the new room. This can be done by multiplying 459 sq ft (the total sq ft of Holden's bedroom and bathroom combined) by 2, which gives us 918 sq ft. Therefore, the new room will have 918 sq ft. The basketball field of Tom is 200 sq ft larger than Holden's bathroom so it is not relevant to the calculation process of this problem. ✓

**I$^3$C-Select**

Holden's bedroom is 309 sq ft and his bathroom is 150 sq ft. The total sq ft of Holden's bedroom and bathroom combined is 459 sq ft. We need to find a space that is twice as large as Holden's bedroom and bathroom combined. Therefore, the new room will be 2 x 459 sq ft = 918 sq ft. The condition The basketball field of Tom is 200 sq ft larger than Holden's bathroom is not relevant to the calculation process of this problem, as it does not involve any calculation involving Holden's bedroom and bathroom. ✓ |

Table 13: Example outputs by Instruct-CoT, Instruct-CoT + I³C and I³C-Select. Irrelevant conditions in the problem description is highlighted with red color, and the LLM-generated final answer is highlighted with yellow color.

| Problem | James buys 6 t-shirts for 50% off. They each cost $20. Emma works at the mall with an hourly salary of $10. How much did James pay? |
|---|---|
| **Gold Answer** | $(20/2) \times 6 = 60$ |
| **Verification Outputs** | ■ First, James paid 50% off the normal price for the 6 t-shirts, which would normally cost $20 each. This means he paid $10 per t-shirt. So, James paid $60 total for the 6 t-shirts. The condition James buys 6 t-shirts for 50% off is relevant to the calculation process, as it determined the amount that James paid for the 6 t-shirts. Emma's hourly salary of $10 is not relevant to the calculation process of this problem. ■ James bought 6 t-shirts, so he paid for 6 times $20. Then, he got 50% off, so 50% of $20 is $10, so he paid for 6 times $10. The condition They each cost $20 is relevant to the problem, because it tells us how much James paid for each t-shirt. So, the answer is James paid $60. ■ James bought 6 t-shirts at 50% off, which means the t-shirts were 50% cheaper than their original price. This means that each t-shirt cost $20. Multiplying 6 t-shirts by $20, James paid a total of $120 for the 6 t-shirts. The condition that Emma works at the mall with an hourly salary of $10 is not relevant to the calculation process of this problem. |
| **Reasoning Path** | **Instruct-CoT** Since James bought 6 t-shirts for 50% off, the price of each t-shirt is reduced to $20. Therefore, James paid $120 in total for the 6 t-shirts. Emma's hourly salary of $10 is irrelevant to the question. ✗ **Instruct-CoT + I³C** James bought 6 t-shirts that were 50% off, so the original price of each t-shirt was $20. Therefore, James paid $10 for each t-shirt, and he bought 6 t-shirts, so he paid $60 in total. The condition that Emma works at the mall with an hourly salary of $10 is not relevant to the calculation process of this problem. ✓ **I³C-Select** First, James paid 50% off the normal price for the 6 t-shirts, which would normally cost $20 each. This means he paid $10 per t-shirt. So, James paid $60 total for the 6 t-shirts. The condition James buys 6 t-shirts for 50% off is relevant to the calculation process, as it determined the amount that James paid for the 6 t-shirts. Emma's hourly salary of $10 is not relevant to the calculation process of this problem. ✓ |

Table 14: Example outputs by Manual-CoT, Manual-CoT + I$^3$C and I$^3$C-Select. Irrelevant conditions in the problem description is highlighted with red color, and the LLM-generated final answer is highlighted with yellow color.

| Problem | James collects all the fruits from his 2 trees. Each tree has 20 plants. Each plant has 1 seed and he plants 60% of those. James' sister is 10 years old. How many trees did James plant? |
|---|---|
| **Gold Answer** | $2 \times 20 \times 60\% = 24$ |
| **Verification Outputs** | ■ James collected all the fruits from his 2 trees. This means he did not plant any new trees. Each tree has 20 plants. This means James had 40 plants in total. Each plant has 1 seed. This means James had 40 seeds in total. He planted 60% of those seeds. This means he planted 24 seeds. Therefore, James planted 24 seeds, but he did not plant any new trees. The condition James collects all the fruits from his 2 trees is relevant to the calculation process because it indicates that James did not plant any new trees, only seeds. 

 ■ First, James collected all the fruits from his two trees. Each tree has 20 plants, so James collected 40 plants in total. Second, each plant has 1 seed, so James collected 40 seeds in total. Third, he planted 60% of those seeds, so James planted 24 seeds in total. Finally, we can conclude that James planted two trees, since each tree had 20 plants and he planted 24 seeds in total. So, to answer the question, James planted two trees. Yes, the condition Each tree has 20 plants is relevant to the calculation process of the problem How many trees did James plant?, since it helps us determine how many plants he collected in total and how many seeds he planted in total. 

 ■ James collects all the fruits from his 2 trees. Each tree has 20 plants. Each plant has 1 seed. This means there are 40 seeds in total. James then plants 60% of those, which means he plants 24 seeds. The condition James' sister is 10 years old is not relevant to the calculation process of this problem. Therefore, James planted 24 trees. |
| **Reasoning Path** | **Manual-CoT** 

 James collected all the fruits from his 2 trees. Each tree has 20 plants. Each plant has 1 seed. He planted 60% of those. 60% is 0.6, so he planted 0.6 x 20 = 12 of those. James' sister is 10 years old. This has nothing to do with the question, so the answer is still 12 . ✗ 

 **Manual-CoT + I$^3$C** 

 James collected all the fruits from his two trees, so he has 40 plants. Each plant has 1 seed, so he has 40 seeds. He plants 60% of those, so he plants 24 seeds. Therefore, James planted 24 trees. The condition James' sister is 10 years old is not relevant to the calculation process of this problem. ✓ 

 **I$^3$C-Select** 

 First, James collected all the fruits from his two trees, so he has 40 plants. Second, each plant has 1 seed, so he has 40 seeds. Third, he plants 60% of those, so he plants 24 seeds. Therefore, James planted 24 trees. The condition that James' sister is 10 years old is not relevant to the calculation process of this problem. The relevant information to answer this question is that James collected all the fruits from his two trees, each tree has 20 plants, and he plants 60% of those. To calculate how many trees James planted, we need to calculate the number of seeds he planted. So the answer would be: 24 trees. ✓ |

Table 15: Example outputs by Auto-CoT, Auto-CoT + I$^3$C and I$^3$C-Select. Irrelevant conditions in the problem description is highlighted with red color, and the LLM-generated final answer is highlighted with yellow color.

| Problem | Hamza has several empty buckets of different sizes, holding either 3, 5, or 6 liters. She fills the 5-liter bucket and pours as much as she can into the 3-liter bucket. Then, she pours the remainder into the 6-liter bucket. Jack drinks 7 liters of milk every day. How much more water, in liters, can Hamza put into the 6-liter bucket, without overflowing? |
|---|---|
| **Gold Answer** | $6 - (5 - 3) = 4$ |
| **Verification Outputs** | ■ First, Hamza fills the 5-liter bucket and then pours as much as she can into the 3-liter bucket. Since the 3-liter bucket has a capacity of 3 liters, she can pour all 5 liters into it. Then, she pours the remainder (2 liters) into the 6-liter bucket. That means she has put a total of 7 liters into the 6-liter bucket. Now, Jack drinks 7 liters of milk every day. That means the 6-liter bucket is emptied every day. Therefore, Hamza can put up to 6 liters more water, in liters, into the 6-liter bucket, without overflowing. Yes, the condition Hamza has several empty buckets of different sizes, holding either 3, 5, or 6 liters. is relevant to the calculation process of the problem. This condition gives us the capacity of each bucket, which is necessary to calculate the amount of water Hamza can put into the 6-liter bucket without overflowing.

■ First, Hamza filled the 5-liter bucket with water. She then poured as much of the water as she could into the 3-liter bucket. Since the 3-liter bucket can hold a maximum of 3 liters of water, Hamza would have been able to pour 2 liters of water from the 5-liter bucket into the 3-liter bucket. The remaining 3 liters of water from the 5-liter bucket would then be poured into the 6-liter bucket. This means that the 6-liter bucket currently holds 3 liters of water. Based on this information, we can calculate that Hamza can put an additional 3 liters of water into the 6-liter bucket, without overflowing. This is because the 6-liter bucket can hold a maximum of 6 liters of water, and it currently holds 3 liters of water. The condition Jack drinks 7 liters of milk every day is not relevant to the calculation process of this problem. |
| **Reasoning Path** | **Auto-CoT**

The 5-liter bucket is filled. She pours as much as she can into the 3-liter bucket. That leaves 2 liters in the 5-liter bucket. She then pours the remainder into the 6-liter bucket. That leaves 2 liters in the 6-liter bucket. Jack drinks 7 liters of milk every day, so she has to leave 7 liters of water for him. Therefore, she can put 5 liters of water in the 6-liter bucket without overflowing. The answer is 5. ✗

**Auto-CoT + I$^3$C**

Hamza first fills the 5-liter bucket. She then pours as much as she can into the 3-liter bucket, filling it to capacity. The remaining 2 liters of water from the 5-liter bucket are then poured into the 6-liter bucket, leaving 4 liters of space for more water (6 liters - 2 liters = 4 liters). Hamza can add 4 more liters of water into the 6-liter bucket without overflowing. The answer is 4. ✓

**I$^3$C-Select**

Hamza has a 5-liter bucket. This is relevant to the calculation process as it tells us the size of the bucket Hamza has. She fills the 5-liter bucket and pours as much as she can into the 3-liter bucket. This is relevant to the calculation process as it tells us how much water Hamza is transferring from the 5-liter bucket to the 3-liter bucket. The 3-liter bucket can only hold 3 liters of water, so Hamza would be able to pour 3 liters of water from the 5-liter bucket into the 3-liter bucket. She then pours the remainder of the water (2 liters) into the 6-liter bucket. This is relevant to the calculation process as it tells us how much water Hamza is transferring from the 5-liter bucket to the 6-liter bucket. The 6-liter bucket currently has 2 liters of water in it, so it can hold 4 more liters of water without overflowing. So the answer would be 4 liters. ✓ |

