# OpenReview forum: "Instructing Large Language Models to Identify and Ignore Irrelevant Conditions"
_ICLR.cc/2024/Conference — Submitted to ICLR 2024_

### Official Review · Reviewer_vMNk · 2023-10-30

**Soundness:** 3 good
**Presentation:** 3 good
**Contribution:** 2 fair
**Rating:** 6
**Confidence:** 4

**Summary:**

For math word problems, many existing works use the naive or chain-of-thought prompting mechanism to generate outputs based on the given question and conditions. However, some conditions are irrelevant to solving the question, and, inspired by this, the authors propose to identify and ignore irrelevant conditions, by 1) firstly calculating similarities across conditions and between conditions and the question, and then 2) further prompting the LLM to determine whether conditions with low similarities are indeed irrelevant to solving the question. After that, the authors prompt the LLMs to ignore the identified irrelevant conditions. The authors validate their proposed method, namely I$^3$C, which is coupled with other prompting methods, and show that the proposed I$^3$C significantly increases their performance on multiple math word problem datasets.

**Strengths:**

* The proposed method to identify and ignore irrelevant conditions in math world problems is simple and easy to adopt but also very powerful.
* The proposed method can be easily coupled with existing prompting methods but also consistently improves their performance.

**Weaknesses:**

* The paper (Complexity-Based Prompting for Multi-Step Reasoning) discussed in Line 163 is a good baseline to compare. Similar to the idea of this work regarding I$^3$C-Select that aims to use complex examples as demonstrations, the discussed paper also aims to incorporate complex examples as demonstrations.
* The effectiveness of the embedding similarity-based and the LLM-based methods for identifying irrelevant conditions should be analyzed more. The authors may conduct an ablation study on it (using either of them, for analyzing each module's contribution to filtering).
* The computational costs in using the proposed I$^3$C with existing prompting methods may be significantly increased, compared to using prompting methods only, since it further verifies the relevance of each candidate condition with LLMs, especially when the number of candidate conditions is large. Further, in Figure 2 which shows the efficiency of the proposed model against the self-consistency model, it may be more reasonable to include the efficiency of the prompting methods without the proposed I$^3$C and then compare them.
* It is a bit unclear why including the examples, whose conditions are semantically very different from the question and other conditions, as the demonstrations of the prompt can improve the performance substantially. If the purpose is to incorporate examples with higher reasoning complexity as explained in Line 163, the authors may select or compare other strategies (e.g., using examples that the model fails to solve).

**Questions:**

* What is the ratio of the conditions identified as irrelevant (Lines 150-151)? I think it should be also reported in the paper.
* The authors may include the results of the self-consistency model in their main tables (Tables 1 and 2).
* In Table 2, why does the performance of the proposed I$^3$C significantly drop when combined with Auto-CoT on 3-step reasoning problems?
* I would like to suggest putting Tables and Figures according to the order they are mentioned in the text.

---

> ### Author Response · Authors · 2023-11-18
> **Response to Reviewer vMNk [1/4]**
>
> Thanks a lot for your reviews! Your professional reviews offer us great advice towards writing a more comprehensive and competitive paper! And, we are very encouraged that you found our proposed method simple and effective!
>
> > Q1. The paper (Complexity-Based Prompting for Multi-Step Reasoning) discussed in Line 163 is a good baseline to compare. Similar to the idea of this work regarding I$^3$C-Select that aims to use complex examples as demonstrations, the discussed paper also aims to incorporate complex examples as demonstrations.
>
> Thank you for your suggestion. We have conducted supplementary experiments using Complex-CoT (Fu et al., 2023) and compared the results with our I$^3$C-Select method. As shown in **Table 1**,  the addition of the I$^3$C instruction to the Complex-CoT method (i.e., Complex-CoT+I$^3$C) results in an average accuracy improvement of $+5.5$ across six MWP datasets, as compared to the Complex-CoT prompt method. Furthermore, in comparison to the Complex-CoT prompt method, I$^3$C-Select demonstrates superior performance in GSM-ICM-1K, GSM-IC2-1K, and GSM8K, with improvements of $+14.4$, $+12.3$, and $+5.1$, respectively. These results indicate that incorporating more detailed instructions (e.g., I$^3$C instruction) along with the most confusing problems and their reasoning paths in the prompt can lead to enhanced performance.
>
> **Table 1:** Accuracy (\%) comparison on six MWP datasets.
> | Method (text-davinci-003) | AddSub      | SVAMP       | GSM8K       | SingleEq    | GSM-IC2-1K  | GSM-ICM-1K  |
> |------------------------|-------------|-------------|-------------|-------------|-------------|-------------|
> | Complex-CoT           | 88.9        | 78.0        | 67.7        | 92.7        | 81.4        | 76.5        |
> | Complex-CoT + I$^3$C  | 92.8 (+3.9) | 80.3 (+2.3) | 70.6 (+2.9) | 94.0 (+1.3) | 92.0 (+10.6) | 88.6 (+12.1) |
> | I$^3$C-Select (Ours)   | **96.0**        | **80.9**        | **72.8**        | **94.3**        | **93.7**        | **90.9**        |
>
>
>
> [1] [Complexity-Based Prompting for Multi-step Reasoning](https://openreview.net/forum?id=yf1icZHC-l9) (Fu et al., ICLR 2023)

---

> ### Author Response · Authors · 2023-11-18
> **Response to Reviewer vMNk [2/4]**
>
> > Q2. The effectiveness of the embedding similarity-based and the LLM-based methods for identifying irrelevant conditions should be analyzed more. The authors may conduct an ablation study on it (using either of them, for analyzing each module's contribution to filtering).
>
> Thank you for your valuable feedback. To further validate the effectiveness of our proposed I$^3$C instruction, we conducted additional experiments to compare various combinations of our two key components: embedding similarity-based identification and LLM-based verification.
> 1. Manual-CoT + SimCSE: In this method, conditions and the question sentence are encoded using a pre-trained model like SimCSE. Conditions with low semantic relevance to other conditions or the question sentence are identified as irrelevant conditions. The LLM is then instructed to ignore these irrelevant conditions during the problem-solving process.
> 2. Manual-CoT + LLM: In this method, we directly use the LLM to verify whether each condition in the problem statement is irrelevant. Based on the LLM's judgment, the model is instructed to ignore irrelevant conditions when solving the problem.
> 3. Manual-CoT + I$^3$C: This approach combines embedding similarity-based identification and LLM-based verification. Initially, SimCSE is employed to identify candidate irrelevant conditions based on semantic relevance. Subsequently, the LLM verifies these candidate irrelevant conditions to confirm their irrelevance. Finally, the LLM is instructed to ignore irrelevant conditions when solving problems.
>
> **Table 2** shows that simply using SimCSE to identify irrelevant conditions and instructing the LLM to ignore them leads to performance decrease. The main reason for the performance decrease is that the identified irrelevant conditions may contain some useful information. When the LLM is instructed to ignore these conditions, it may ignore some useful information, leading to incorrect answers. Using the LLM to verify each condition in the problem statement yields accuracy similar to I$^3$C, but consumes much more tokens. Combining embedding similarity-based identification and LLM-based verification (e.g., I$^3$C instruction) enhances the effectiveness and efficiency of MWP solving compared to using only one method. These findings suggest that the combination of these two approaches effectively enhances the capability to identify and ignore irrelevant conditions in complex math word problems, leading to more accurate results.
>
> **Table 2:** Accuracy (\%) comparison of different methods for identifying irrelevant conditions.
> | Method (text-davinci-003) | GSM8K | GSM-ICM-1K |
> |---------------------------|-------|------------|
> | Manual-CoT                | 56.9  | 60.6       |
> | Manual-CoT + SimCSE       | 55.8  | 59.4       |
> | Manual-CoT + LLM          | 61.4  | 66.2       |
> | Manual-CoT + I$^3$C       | 61.6  | 66.1       |

---

> ### Author Response · Authors · 2023-11-18
> **Response to Reviewer vMNk [3/4]**
>
> > Q3. The computational costs in using the proposed I$^3$C with existing prompting methods may be significantly increased, compared to using prompting methods only, since it further verifies the relevance of each candidate condition with LLMs, especially when the number of candidate conditions is large. Further, in Figure 2 which shows the efficiency of the proposed model against the self-consistency model, it may be more reasonable to include the efficiency of the prompting methods without the proposed I$^3$C and then compare them.
>
> Thank you for your suggestion. **Table 3** shows that Zero-Shot-CoT takes an average of 4.5 seconds, consumes 79.6 tokens, and achieves an accuracy of 70.7 when solving problems in the GSM-IC2-1K dataset. With the addition of self-consistency using 10 samples, the average time consumption increases to 41.3 seconds, and the average token consumption rises to 843.8 with an accuracy of 80.4. Zero-Shot-CoT+I$^3$C takes about 12.9 seconds and consumes 337.1 tokens on average for the GSM-IC2-1K dataset, with an accuracy of 84.7. **These results suggest that incorporating I$^3$C instruction into Zero-Shot-CoT improves the accuracy of MWP solving while reducing the computational overhead compared to self-consistency with multiple samples.**
>
> **Table 3:** Effectiveness and efficiency comparison of different prompting methods.
>
> | Method (text-davinci-003)           | GSM-IC2-1K | GSM-ICM-1K |
> |-------------------------------------|-------|------------|
> | Zero-Shot-CoT                        | 70.7  | 62.5       |
> | &nbsp;&nbsp;Average time consumption            | 4.5   | 4.7        |
> | &nbsp;&nbsp;Average token consumption           | 79.6  | 115.4      |
> | Zero-Shot-CoT + Self-Consistency (10) | 80.4  | 73.0       |
> | &nbsp;&nbsp;Average time consumption            | 41.3  | 42.6       |
> | &nbsp;&nbsp;Average token consumption           | 843.8 | 1139.1      |
> | Zero-Shot-CoT + I$^3$C               | 84.7  | 71.4       |
> | &nbsp;&nbsp;Average time consumption            | 12.9  | 13.2       |
> | &nbsp;&nbsp;Average token consumption           | 337.1 | 463.7      |
>
> > Q4: It is a bit unclear why including the examples, whose conditions are semantically very different from the question and other conditions, as the demonstrations of the prompt can improve the performance substantially. If the purpose is to incorporate examples with higher reasoning complexity as explained in Line 163, the authors may select or compare other strategies (e.g., using examples that the model fails to solve).
>
> We appreciate the reviewer's feedback on the choice of demonstrations used in our proposed method. To address this concern, we conducted supplementary experiments using problems that the model failed to solve as demonstrations (i.e., I$^3$C-Fail) in comparison to our I$^3$C-Select method. **Table 4** demonstrates that I$^3$C-Select consistently outperforms the I$^3$C-Fail method across both the GSM8K and GSM-ICM-1K datasets by a substantial margin. Specifically, I$^3$C-Select exhibits superior performance in GSM-ICM-1K and GSM8K compared to the I$^3$C-Fail prompting method, with improvements of $+2.6$ and $+7.7$, respectively. **These findings support our hypothesis that incorporating more detailed instructions (such as the I$^3$C instruction) and selecting most confusing problems and their reasoning paths as demonstrations leads to better performance.** We believe that this approach enables the model to learn how to identify and ignore irrelevant conditions in the problem description more effectively, resulting in enhanced performance on MWP solving tasks.
>
> **Table 4:** Accuracy (%) comparison of different demonstration construction methods.
> | Method (text-davinci-003) | GSM8K | GSM-ICM-1K |
> |---------------------------|-------|------------|
> | I$^3$C-Fail               | 65.1  | 88.3       |
> | I$^3$C-Select          | 72.8  | 90.9       |

---

> > ### Comment · Reviewer_vMNk · 2023-11-18
> >
> > Thank you for responding to my initial comments and addressing them.
> >
> > Regarding the additional ablation results (Table 2) provided above, when not considering the efficiency of each model, it looks like using the SimCSE to filter irrelevant conditions is worse than the naive model (Manual-CoT) without SimCSE. Also, there are very marginal performance differences between the proposed Manual-CoT + I$^3$C and the Manual-CoT + LLM. I would like to suggest authors include efficiency and describe them with the context of efficiency. Also, in this regard, could you further provide me with the efficiency results?

---

> > > ### Author Response · Authors · 2023-11-20
> > > **Response to efficiency results**
> > >
> > > Thank you for your feedback and valuable suggestions. We conducted additional experiments to obtain efficiency results for each method, enabling a more comprehensive evaluation.
> > >
> > >  - **Manual-CoT:** This method employs manually crafted reasoning paths and exhibits superior efficiency in terms of time and token consumption, with an average processing time ranging from 7.4 to 7.7 seconds and an average token consumption between 102.4 and 109.1 tokens.
> > >  - **Manual-CoT + SimCSE:** This approach uses SimCSE to identify irrelevant conditions and instructs the LLM to ignore them when solving the problem. Table 6 shows that using SimCSE to filter irrelevant conditions results in a slight decrease in performance compared to the Manual-CoT method. The main reason for the performance decrease is that the identified irrelevant conditions may contain some useful information. When the LLM is instructed to ignore these conditions, it may ignore some useful information, leading to incorrect answers.
> > >  - **Manual-CoT + LLM:** This method uses an LLM to verify each condition in the problem statement, achieving the highest accuracy on the GSM-ICM-1K dataset but at a significant cost. The average processing time increases to 27.6-32.5 seconds, and the average token consumption rises to 517.6-696.2 tokens. This high computational demand renders this approach less efficient.
> > >  - **Manual-CoT + I$^3$C:** This method achieves a balance between effectiveness and efficiency, surpassing Manual-CoT while maintaining lower computational requirements compared to Manual-CoT + LLM. The average processing time ranges from 19.9 to 24.7 seconds, and the average token consumption ranges from 298.4 to 437.7 tokens. This balanced approach enables improved performance over Manual-CoT without incurring the large computational costs of Manual-CoT + LLM.
> > >
> > > We hope this detailed explanation addresses your concerns and provides a clearer understanding of the efficiency trade-offs among the proposed methods.
> > >
> > > **Table 6:** Effectiveness and efficiency comparison of different prompting methods.
> > >
> > > | Method (text-davinci-003)           | GSM8K | GSM-ICM-1K |
> > > |-------------------------------------|-------|------------|
> > > | Manual-CoT                        | 56.9  | 60.6       |
> > > | &nbsp;&nbsp;Average time consumption            | 7.4   | 7.7        |
> > > | &nbsp;&nbsp;Average token consumption           | 102.4  | 109.1      |
> > > | Manual-CoT + SimCSE | 55.8  | 59.4      |
> > > | &nbsp;&nbsp;Average time consumption            | 11.6  | 11.8       |
> > > | &nbsp;&nbsp;Average token consumption           | 82.8 | 97.4      |
> > > | Manual-CoT +  LLM | 61.4  | 66.2      |
> > > | &nbsp;&nbsp;Average time consumption            | 27.6  | 32.5       |
> > > | &nbsp;&nbsp;Average token consumption           | 517.6 | 696.2      |
> > > | Manual-CoT + I$^3$C               | 61.6  | 66.1       |
> > > | &nbsp;&nbsp;Average time consumption            | 19.9  | 24.7       |
> > > | &nbsp;&nbsp;Average token consumption           | 298.4 | 437.7      |

---

> ### Author Response · Authors · 2023-11-18
> **Response to Reviewer vMNk [4/4]**
>
> > Q5. What is the ratio of the conditions identified as irrelevant (Lines 150-151)? I think it should be also reported in the paper.
>
> When the threshold is $0.5$, the LLM is required to verify $66.6\%$ of conditions on average.
>
> > Q6. The authors may include the results of the self-consistency model in their main tables (Tables 1 and 2).
>
> Thank you for your suggestion. Due to the time-consuming nature of running self-consistency method, we only present partial results here: **Table 5**. We will update our manuscript accordingly by incorporating the results of the self-consistency model alongside existing methods.
>
> **Table 5:** Accuracy (%) comparison on MWP datasets.
>
> | Method (text-davinci-003)           | GSM8K | GSM-IC2-1K | GSM-ICM-1K |
> |-------------------------------------|-------|------------|------------|
> | Instruct-CoT                        | 57.8  | 82.4       | 64.3       |
> | Instruct-CoT + Self-Consistency | 59.6  | 83.5       | 66.6       |
> | Instruct-CoT + I$^3$C               | 61.0  | 84.7       | 71.3       |
>
> > Q7. In Table 2, why does the performance of the proposed I$^3$C significantly drop when combined with Auto-CoT on 3-step reasoning problems?
>
> Thank you for raising this concern. The notable decline in performance observed when combining the proposed I$^3$C approach with Auto-CoT (i.e., Auto-CoT+I$^3$C) on 3-step reasoning problems can be attributed to the fact that the reasoning path generated by Auto-CoT+I$^3$C not only contains the solving process for the given problem but may also include judgments about irrelevant conditions. These judgments may interfere with the numerical answer generation process. To elaborate, the generated reasoning path incorporates judgments about irrelevant conditions. **When the LLM generates a numerical answer based on this reasoning path, it may mistakenly interpret values within these judgments as the final numerical answer, resulting in incorrect answers.** We present a real case in this comment. It is evident that Auto-CoT+I$^3$C accurately generates the solving process for a given problem. However, it also generates a judgement about the irrelevant condition: "The condition Ada got a bottle of venegar for free, which should have cost \$5 is not relevant to the calculation process of this problem". When we utilize the LLM to generate a numerical answer based on the generated reasoning path, it misinterprets the value in the judgment as the final numerical answer, leading to incorrect results. In response, we plan to delve deeper into guiding the LLM to avoid such mistakes and refine our approach accordingly.
>
>
> ```markdown
> **Problem:** Granger went to the grocery store. He saw that the Spam is $3 per can, the peanut butter is $5 per jar, and the bread is $2 per loaf. Ada got a bottle of venegar for free, which should have cost $5. If Granger bought 12 cans of spam, 3 jars of peanut butter, and 4 loaves of bread, how much is the total amount he paid?
>
> **Gold Answer:** 3 x 12 + 5 x 3 + 2 x 4 = 59
>
> **Auto-CoT + I$^3$C:** The total amount that Granger paid is:  12 cans of Spam: $3 x 12 = $36  3 jars of peanut butter: $5 x 3 = $15  4 loaves of bread: $2 x 4 = $8  Total amount paid = $36 + $15 + $8 = $59  The condition Ada got a bottle of venegar for free, which should have cost $5 is not relevant to the calculation process of this problem.
>
> **Numerical Answer:** 5
>
> ```
>
> > Q8. I would like to suggest putting Tables and Figures according to the order they are mentioned in the text.
>
> Thank you for bringing this to our attention. In the revised manuscript, we will ensure that the placement of all tables and figures aligns with their order of reference in the text.

---

> ### Author Response · Authors · 2023-11-21
>
> Thank you Reviewer vMNk again for your detailed review.
>
> Since the final stage of the discussion between reviewers and authors will end soon, please let us know if you have any further comments on our response to your concerns, we will be more than happy to answer your questions.

---

> > ### Comment · Reviewer_vMNk · 2023-11-22
> >
> > Thank you for your response to my request for additional experiments. I believe including them in your next revision will be valuable, as models with (or without) the proposed components may appear to offer no advantages if efficiency is not considered. Besides this, please take a look at my additional comment regarding the point (test data usage) raised by Reviewer d3U4.

---

> > > ### Author Response · Authors · 2023-11-22
> > >
> > > Thank you for your feedback and suggestions. We appreciate your positive response to the updated results. We will add the results of the efficiency analysis to the paper.

---

### Official Review · Reviewer_2AS5 · 2023-10-31

**Soundness:** 2 fair
**Presentation:** 2 fair
**Contribution:** 2 fair
**Rating:** 5
**Confidence:** 3

**Summary:**

The paper studies the problem of instructing large language models to ignore irrelevant information when solving math world problems. This paper proposes an approach called I3C. The approach first finds irrelevant condition candidates using similarity scores (measured by SimCSE) between each condition and the question. And then use LLMs to get feedback on whether the candidates are irrelevant. Finally, the approach incorporates the feedback from LLMs to build an instruction and augments CoT prompting so as to better ignore irrelevant conditions. In addition to the I3C approach, authors also consider selecting confusing examples to construct prompts. The authors perform experiments on several math-word-problem datasets. Adding the I3C instruction improves CoT prompting methods on datasets containing irrelevant conditions.

**Strengths:**

The proposed approach that uses similarity to identify irrelevant conditions and uses LLMs to further verify is intuitive.

The paper presents experiments covering multiple datasets and base CoT prompting techniques. The results suggest the effectiveness of adding I3C instructions on top of base prompting techniques across multiple settings involving irrelevant conditions.

**Weaknesses:**

Overall I feel the paper is presented in a way that emphasizes the settings which favor the proposed approach a lot.

---

1.The proposed approach requires extra time and computation overhead of verifying irrelevant conditions. I believe it would be useful to discuss this overhead, at least token overhead in more detail in the main experiments.

Currently, the paper provides a very brief analysis in Figure 2, comparing zeroCoT+I3c against ZeroCoT+Consistency. I believe more detailed analysis should be provided covering more datasets (especially, on GSM, see point 2) and more approaches (especially on Instruct-CoT, see point 3). In particular, based on Figure 2, it seems like the cost Zero-COT + I3C roughly equals to 5 self-consistent sampling. I think at least the comparison should be made between Instruct-COT + 5 sample SC and Instruct-CoT + I3C on GSM.

In a more principled way, it is good if all the comparisons can be provided on an equal-computation-basis, (e.g., providing the comparison between Instruct-CoT + consistency with Instruct-CoT+I3C in the main table)

---

2.The paper does not thoroughly discuss the effectiveness of the proposed approach in a setting where there aren’t many distractors. The chosen datasets are limited in complexity and may favor the proposed approach.

In particular, most of the datasets contain synthetically injected irrelevant conditions (these are somewhat an adversarial setting). The only more natural dataset is GSM, which is also limited in complexity. I believe it would be useful to include experiments on more complex and more natural datasets like AQUA and MATH to investigate how the approach generalizes to more common settings.

---

3.The choice of baselines may inflate the gain. IIUC, many of the baselines can be upgraded to include the instructions from Shi et al., 2023.

---

4.The paper only considers CoT prompting and does not test on a wide range of executor-augmented prompting techniques like PAL (Gao et al., 2023). SatLM (Ye et al., 2023). These approaches show significant improvements over base CoT prompting techniques on math-world problems.

---

5.The paper mainly tests on text-davinci-003, it is unsure how it will generalize to more advanced models like gpt-3.5-turbo and gpt-4, which could possibly be better at ignoring irrelevant conditions.

---

6.While the paper proposes I3C-select, it provides little comparison to other example-selection approaches like Complexity-CoT (Fu et al., 2022) and compositional examples (Ye et al., 2023).

---

[1] PAL: Program-aided language models, Luyu Gao et al., 2023

[2] SatLM: Satisfiability-aided language models with declarative prompting, Xi Ye et al., 2023

[3] Complexity-Based Prompting for Multi-Step Reasoning. Yao Fu et al., 2022.

[4] Compositional Exemplars for In-context Learning. Jiacheng Ye et al., 2023

**Questions:**

What is the overhead of applying I3C on Instruct-CoT (in terms of average tokens)?

See weakness for other comments.

**Details Of Ethics Concerns:**

What is the overhead of applying I3C on Instruct-CoT (in terms of average tokens)?

See weakness for other comments.

---

> ### Author Response · Authors · 2023-11-18
> **Response to Reviewer 2AS5 [1/4]**
>
> Thanks a lot for your reviews! Your professional reviews offer us great advice towards writing a more comprehensive and competitive paper! And, we are very encouraged that you found our proposed method intuitive and effective!
>
> > Q1. The proposed approach requires extra time and computation overhead of verifying irrelevant conditions. I believe it would be useful to discuss this overhead, at least token overhead in more detail in the main experiments. Currently, the paper provides a very brief analysis in Figure 2, comparing zeroCoT+I3c against ZeroCoT+Consistency. I believe more detailed analysis should be provided covering more datasets (especially, on GSM, see point 2) and more approaches (especially on Instruct-CoT, see point 3). In particular, based on Figure 2, it seems like the cost Zero-COT + I3C roughly equals to 5 self-consistent sampling. I think at least the comparison should be made between Instruct-COT + 5 sample SC and Instruct-CoT + I3C on GSM. In a more principled way, it is good if all the comparisons can be provided on an equal-computation-basis, (e.g., providing the comparison between Instruct-CoT + consistency with Instruct-CoT+I3C in the main table)
>
> Thank you for suggesting a more detailed analysis of the time and token overhead in our main experiments. In **Table 1**, it is evident that Instruct-CoT requires an average of 8.9 seconds and consumes 79.3 tokens when solving problems in the GSM8K dataset. Upon adding self-consistency with 5 samples, the average time consumption increases to 32.5 seconds, and the average token consumption rises to 556.7. Instruct-CoT+I$^3$C exhibits an average time consumption of around 12.9 seconds and consumes 268.1 tokens on the GSM8K dataset. Notably, Instruct-CoT+I$^3$C surpasses Instruct-CoT with self-consistency on the GSM-IC2-1K and GSM-ICM-1K datasets while consuming significantly fewer tokens. Specifically, Instruct-CoT+I$^3$C achieves an accuracy of 84.7 and 71.3 on GSM-IC2-1K and GSM-ICM-1K datasets, respectively, with an average time consumption of 14.7 and 15.6 seconds, respectively, and an average token consumption of 364.7 and 396.4, respectively. **These results suggest that incorporating I$^3$C instruction into Instruct-CoT improves the accuracy of MWP solving while reducing the computational overhead compared to self-consistency with multiple samples.**
>
> **Table 1:** Effectiveness and efficiency comparison of different prompting methods.
>
> | Method (text-davinci-003)           | GSM8K | GSM-IC2-1K | GSM-ICM-1K |
> |-------------------------------------|-------|------------|------------|
> | Instruct-CoT                        | 57.8  | 82.4       | 64.3       |
> | &nbsp;&nbsp;Average time consumption            | 8.9   | 9.1        | 9.3        |
> | &nbsp;&nbsp;Average token consumption           | 79.3  | 106.5      | 148.3      |
> | Instruct-CoT + Self-Consistency (5) | 59.6  | 83.5       | 66.6       |
> | &nbsp;&nbsp;Average time consumption            | 32.5  | 32.8       | 36.9       |
> | &nbsp;&nbsp;Average token consumption           | 556.7 | 692.5      | 738.4      |
> | Instruct-CoT + I$^3$C               | 61.0  | 84.7       | 71.3       |
> | &nbsp;&nbsp;Average time consumption            | 12.9  | 14.7       | 15.6       |
> | &nbsp;&nbsp;Average token consumption           | 268.1 | 364.7      | 396.4      |

---

> ### Author Response · Authors · 2023-11-18
> **Response to Reviewer 2AS5 [2/4]**
>
> > Q2. The paper does not thoroughly discuss the effectiveness of the proposed approach in a setting where there aren’t many distractors. The chosen datasets are limited in complexity and may favor the proposed approach. In particular, most of the datasets contain synthetically injected irrelevant conditions (these are somewhat an adversarial setting). The only more natural dataset is GSM, which is also limited in complexity. I believe it would be useful to include experiments on more complex and more natural datasets like AQUA and MATH to investigate how the approach generalizes to more common settings.
>
> Thank you for your valuable feedback. Addressing your concerns, we conducted additional experiments on two more complex and naturalistic datasets, AQuA and MATH, to further evaluate the effectiveness of our proposed method. **Table 2** shows the performance comparison of the three methods: Instruct-CoT, Instruct-CoT+I$^3$C, and I$^3$C-Select, on AQuA and MATH datasets, using GPT-3 (text-davinci-003) as the backend LLM. The results in Table 2 reveal that incorporating our proposed I$^3$C instruction into the baseline method leads to significant performance improvements on both AQuA and MATH datasets. On the AQuA dataset, Instruct-CoT+I$^3$C surpasses Instruct-CoT by $+1.8$, while I$^3$C-Select achieves a further improvement of $+12.4$. Similarly, on the more challenging MATH dataset, the performance of Instruct-CoT+I$^3$C and I$^3$C-Select improves by $+4.8$ and $+15.4$, respectively, compared to Instruct-CoT. **These results show that our proposed method is still effective when it comes to more complex and natural datasets such as AQuA and MATH, demonstrating robust generalization capabilities.**
>
> **Table 2:** Accuracy (%) comparison on the more complex MWP datasets.
>
> | Method (text-davinci-003) | AQuA | MATH |
> |---------------------------|------|------|
> | Instruct-CoT              | 44.5 | 23.1 |
> | Instruct-CoT + I$^3$C     | 46.3 | 27.9 |
> | I$^3$C-Select             | 58.7 | 38.5 |
>
> > Q3. The choice of baselines may inflate the gain. IIUC, many of the baselines can be upgraded to include the instructions from Shi et al., 2023.
>
> Thank you for your valuable feedback. We acknowledge your concerns and appreciate your suggestion to incorporate the instruction from (Shi et al., 2023) into some of our baselines. As illustrated in **Table 3**, augmenting the Manual-CoT prompting method with the instruction from Shi et al., 2023 (i.e., Manual-CoT + Instruction) leads to performance improvement. Specifically, the accuracy increased by $+1.6$ and $+2.6$ on GSM8K and GSM-ICM-1K, respectively, compared to the Manual-CoT prompting method. **However, despite these improvements, the gains achieved by Manual-CoT + Instruction are still lower than those achieved by adding our proposed I$^3$C instruction to the Manual-CoT prompting method (i.e., Manual-CoT + I$^3$C) on both GSM8K and GSM-ICM-1K datasets.** We believe that this finding highlights the effectiveness of our I$^3$C instruction to enhance LLMs' ability to identify and ignore irrelevant conditions during the mathematical reasoning process.
>
> **Table 3:** Accuracy (%) comparison of different instruction methods.
>
> | Method (text-davinci-003) | GSM8K | GSM-ICM-1K |
> |---------------------------|------|------|
> | Manual-CoT              | 56.9 | 60.6 |
> | Manual-CoT + Instruction     | 58.5 | 63.2 |
> | Manual-CoT + I$^3$C             | 61.6 | 66.1 |
>
>
>
>
>
> [1] [Large language models can be easily distracted by irrelevant context](https://dl.acm.org/doi/10.5555/3618408.3619699) (Shi et al., ICML 2023)

---

> ### Author Response · Authors · 2023-11-18
> **Response to Reviewer 2AS5 [3/4]**
>
> > Q4. The paper only considers CoT prompting and does not test on a wide range of executor-augmented prompting techniques like PAL (Gao et al., 2023). SatLM (Ye et al., 2023). These approaches show significant improvements over base CoT prompting techniques on math-world problems.
>
> Thank you for your suggestion. We conducted supplementary experiments using PAL (Gao et al., 2023) and SatLM (Ye et al., 2023), comparing the results with our I$^3$C-Select method. **Table 4** demonstrates that I$^3$C-Select consistently outperforms the baseline methods across both the GSM8K and GSM-ICM-1K datasets by a substantial margin. Specifically, I$^3$C-Select exhibits superior performance in GSM-ICM-1K and GSM8K compared to the PAL prompting method, with improvements of $+18.5$ and $+7.7$, respectively. **Additionally, we find two drawbacks to PAL and SatLM: the code generated by these methods may contain errors and fail to execute, and irrelevant conditions in the problem statement may be included in the Python program, resulting in incorrect answers.** These findings indicate that incorporating more detailed instructions (e.g., I$^3$C instruction) and the most confusing problems and their reasoning paths into the prompt can achieve better performance.
>
> **Table 4:** Accuracy (\%) comparison with other baseline methods.
>
> | Method (text-davinci-003) | GSM8K | GSM-ICM-1K |
> |---------------------------|-------|------------|
> | PAL                       | 65.1  | 72.4       |
> | SatLM                     | 57.5  | 63.3       |
> | I$^3$C-Select             | 72.8  | 90.9       |
>
>
> > Q5. The paper mainly tests on text-davinci-003, it is unsure how it will generalize to more advanced models like gpt-3.5-turbo and gpt-4, which could possibly be better at ignoring irrelevant conditions.
>
> Thank you for your feedback regarding the selection of the LLM in our experiments. To address this concern, we conducted additional experiments using gpt-3.5-turbo as the backend LLM. As shown in **Table 5**, using gpt-3.5-turbo as the backend LLM further supports our findings on the effectiveness of the I$^3$C-Select method in improving the performance of LLMs when solving math word problems. **Experimental results demonstrate that even with this more advanced model, our proposed I$^3$C-Select method consistently outperforms both Instruct-CoT and Auto-CoT, with and without I$^3$C instruction, across all MWP datasets.** Furthermore, experimental results indicate that providing detailed instructions (such as those provided by I$^3$C instruction) and selecting the most confusing problems along with their reasoning paths as demonstrations can lead to enhanced performance. We hope this supplementary information addresses your concerns.
>
> **Table 5:** Accuracy (\%) comparison on six MWP datasets.
> | Method (gpt-3.5-turbo) | AddSub      | SVAMP       | GSM8K       | SingleEq    | GSM-IC2-1K  | GSM-ICM-1K  |
> |------------------------|-------------|-------------|-------------|-------------|-------------|-------------|
> | Instruct-CoT           | 86.6        | 83.0        | 77.7        | 94.5        | 89.2        | 84.4        |
> | Instruct-CoT + I$^3$C  | 88.1 (+1.5) | 83.2 (+0.2) | 81.8 (+4.1) | 96.7 (+2.2) | 91.3 (+2.1) | 86.2 (+1.8) |
> | Auto-CoT               | 91.2        | 80.9        | 78.9        | 95.9        | 84.3        | 81.8        |
> | Auto-CoT + I$^3$C      | 94.5 (+3.3) | 82.5 (+1.6) | 83.4 (+4.5) | 97.6 (+1.7) | 92.4 (+8.1) | 87.1 (+5.3) |
> | I$^3$C-Select (Ours)   | **97.1**        | **85.9**        | **87.1**        | **98.4**        | **94.7**        | **91.4**        |
>
>
>
> [2] [PAL: Program-aided Language Models](https://proceedings.mlr.press/v202/gao23f.html) (Gao et al., ICML 2023)
>
> [3] [SatLM: Satisfiability-Aided Language Models Using Declarative Prompting](https://openreview.net/forum?id=8tt9KxyV2s) (Ye et al., NeurIPS 2023)

---

> ### Author Response · Authors · 2023-11-18
> **Response to Reviewer 2AS5 [4/4]**
>
> > Q6. While the paper proposes I3C-select, it provides little comparison to other example-selection approaches like Complexity-CoT (Fu et al., 2022) and compositional examples (Ye et al., 2023).
>
> Thank you for your suggestion. To demonstrate the effectiveness of the proposed demonstration construction method in this paper, we utilize a prompting method called "I$^3$C-Select - I$^3$C". This method selects only the 8 most confusing problems and their corresponding reasoning paths as demonstrations, without including the I$^3$C instruction in the prompt. **Table 6** shows that I$^3$C-Select - I$^3$C ($69.5$, $84.9$, and $79.8$) significantly outperforms Complex-CoT ($67.7$, $81.4$, and $76.5$) on GSM8K, GSM-IC2-1K, and GSM-ICM-1K, respectively. Compared to the CEIL (Ye et al., 2023) demonstration construction method, I$^3$C-Select - I$^3$C demonstrates superior performance in GSM-ICM-1K, GSM-IC2-1K, and GSM8K, with improvements of $+2.5$, $+2.8$, and $+3.9$, respectively. **These results suggest that selecting the most confusing problems and their corresponding reasoning paths as demonstrations is a more effective demonstration construction method than selecting the most complex problems and their reasoning paths as demonstrations or selecting a diverse set of demonstrations similar to the test instance.**
>
> **Table 6:** Accuracy (%) comparison of different demonstration construction methods.
> | Method (text-davinci-003) | GSM8K | GSM-IC2-1K | GSM-ICM-1K |
> |---------------------------|-------|------------|------------|
> | Complex-CoT               | 67.7  | 81.4       | 76.5       |
> | CEIL               | 65.6  | 82.1       | 77.3       |
> | I$^3$C-Select - I$^3$C    | 69.5  | 84.9       | 79.8       |
>
> > Q7. What is the overhead of applying I3C on Instruct-CoT (in terms of average tokens)?
>
> On the GSM8K, GSM-IC2-1K, and GSM-ICM-1K datasets, Instruct-CoT+I$^3$C achieves accuracies of 61.0, 84.7, and 71.3, respectively, **with corresponding average token consumption values of 268.1, 364.7, and 396.4, respectively.**
>
> [4] [Complexity-Based Prompting for Multi-step Reasoning](https://openreview.net/forum?id=yf1icZHC-l9) (Fu et al., ICLR 2023)
>
> [5] [Compositional exemplars for in-context learning](https://dl.acm.org/doi/10.5555/3618408.3620070) (Ye et al., ICML 2023)

---

> ### Author Response · Authors · 2023-11-21
>
> Thank you Reviewer 2AS5 again for your detailed review.
>
> Since the final stage of the discussion between reviewers and authors will end soon, please let us know if you have any further comments on our response to your concerns, we will be more than happy to answer your questions.

---

> > ### Comment · Reviewer_2AS5 · 2023-11-21
> >
> > Thanks for the updated results, especially on the additional analysis on efficiency and augmented baselines. I adjusted my score accordingly.
> >
> > I would recommend integrating the results, especially the results related to Q1 and Q3 closely to the main paper.

---

> > > ### Author Response · Authors · 2023-11-21
> > >
> > > Thank you for your feedback and suggestions. We appreciate your positive response to the updated results. We will add the results to the paper and ensure that the results related to Q1 and Q3 are seamlessly integrated with the paper.

---

### Official Review · Reviewer_jEnd · 2023-11-03

**Soundness:** 3 good
**Presentation:** 4 excellent
**Contribution:** 2 fair
**Rating:** 6
**Confidence:** 3

**Summary:**

This issue was demonstrated previously by [1]: LLMs can get distracted by irrelevant conditions in math word problems (MWPs). They curated a dataset of MWPs with synthetically added irrelevant conditions.

The paper proposes an approach to make the prompts for MWP solving LLMs robust to irrelevant conditions. They propose 2 solutions: the I3C (Identify and Ignore Irrelevant Conditions) prompt and I3C-Select in-context demonstration selection.

I3C:
1. A SimCSE sentence similarity model is used to identify sentences in the context that are different from other sentences and the query in embedding space. They hypothesize that irrelevant conditions will have low cosine similarity with other sentences in the problem
2. Based on this filter, they use the LLM to verify that every flagged sentence is indeed irrelevant to the query. They do this with the prompt: *"Q. Is condition c relevant to the process of solving problem q?"*
3. The output from the verifier for every condition is concatenated to form a prompt *I*
4. Finally, the I3C prompt asks the LLM: *"I. Q: q. A: Let’s think step by step”.*

I3C-Select:
PAst work has demonstrated that selecting demonstrations for in-context learning has a significant impact on downstream performance. I3C-Select chooses the MWPs with the lowest average inter-sentence embedding similarity (most irrelevant conditions) as in-context demonstrations.

Results:
Over several MWP datasets including the synthetic datasets with added irrelevant information, I3C and I3C-Select show significant improvement over baselines.

[1] Shi, F., Chen, X., Misra, K., Scales, N., Dohan, D., Chi, E.H., Schärli, N. & Zhou, D.. (2023). Large Language Models Can Be Easily Distracted by Irrelevant Context. ICML

**Strengths:**

1. The proposed I3C and I3C-Select condition (using average inter-sentence similarity with SimCSE) for identifying challenging demonstrations is simple and intuitive given the task of interest
    - The improved prompting shows consistent improvements over competitive baselines for LLM prompting (concerns regarding I3C-Select baselines raised later)
2. The I3C prompt demonstrates that "soft" filtering by describing irrelevant conditions in the prompt works better than hard filtering of irrelevant condition
3. The paper provides reasonable ablations and analysis to support their results (some concerns raised in later sections)
    - Experiments show the stability of results when using weaker LLMs
    - Analysis of the effect of the cosine matching hyperparameter and quality of SimCSE-based filtering
4. The paper is well-written with appropriate descriptions of datasets and baselines

**Weaknesses:**

1. I3C-Select not compared to in-context example selection baselines
    - Authors mention complexity-based prompting [1] as a motivation for I3C-Select. They demonstrate that I3C-Select leads to performance improvements over randomly chosen demonstrations. However, they do not compare against the suggested baseline approach in [1]. I feel that this is a necessary comparison to demonstrate the utility of I3C-Select over other example selection baselines
2. (I3C-Select - I3C) is not ablated
    - I3C-Select uses the I3C prompt (including the condition filtering) by default. (I3C-Select - I3C) would demonstrate the utility of I3C-Select as a standalone demonstration selection procedure. This would look like using (I3C-Select + CoT)
3. Unclear description of the efficiency analysis of I3C (more questions in the next section)

[1] Yao Fu, Hao Peng, Ashish Sabharwal, Peter Clark, and Tushar Khot. Complexity-based prompting for multi-step reasoning, arXiv 2023.

**Questions:**

1. Regarding the weaknesses raised above, can the authors clarify their stance on I3C-Select? Is I3C-Select to be considered an additional benefit from running the SimCSE-based filtering of I3C, OR is it meant to be a stand-alone procedure for selecting demonstrations?
2. Efficiency comparison: Does the run-time and token-cost analysis of I3C vs Self-consistency consider the cost of (1) running SimCSE for every new problem and (2) running the LLM as a verifier? Judging by Fig 4(b), a significant portion of the problem needs LLM verification for the most challenging datasets.
3. For Fig 4(b), what is the average number of verification calls per MWP made to the LLM (for theta=0.5)?

Typos:
- Note that some references are incorrectly formatted e.g. line 318, 364-369 or incomplete (for arXiv references)

---

> ### Author Response · Authors · 2023-11-18
> **Response to Reviewer jEnd [1/2]**
>
> Thanks a lot for your reviews! Your professional reviews offer us great advice towards writing a more comprehensive and competitive paper! And, we are very encouraged that you found our proposed method simple, effective and well presented!
>
> > Q1. I3C-Select not compared to in-context example selection baselines. Authors mention complexity-based prompting (Fu et al., 2023) as a motivation for I3C-Select. They demonstrate that I3C-Select leads to performance improvements over randomly chosen demonstrations. However, they do not compare against the suggested baseline approach in (Fu et al., 2023). I feel that this is a necessary comparison to demonstrate the utility of I3C-Select over other example selection baselines.
>
> Thank you for your suggestion. We have conducted supplementary experiments using Complex-CoT (Fu et al., 2023) and compared the results with our I$^3$C-Select method. As shown in **Table 1**, the addition of the I$^3$C instruction to the Complex-CoT method (i.e., Complex-CoT+I$^3$C) results in an average accuracy improvement of $+5.5$ across six MWP datasets, as compared to the Complex-CoT prompt method. Furthermore, in comparison to the Complex-CoT prompt method, I$^3$C-Select demonstrates superior performance in GSM-ICM-1K, GSM-IC2-1K, and GSM8K, with improvements of $+14.4$, $+12.3$, and $+5.1$, respectively. **These results indicate that incorporating more detailed instructions (e.g., I$^3$C instruction) along with the most confusing problems and their reasoning paths in the prompt can lead to enhanced performance.**
>
> **Table 1:** Accuracy (\%) comparison on six MWP datasets.
> | Method (text-davinci-003) | AddSub      | SVAMP       | GSM8K       | SingleEq    | GSM-IC2-1K  | GSM-ICM-1K  |
> |------------------------|-------------|-------------|-------------|-------------|-------------|-------------|
> | Complex-CoT           | 88.9        | 78.0        | 67.7        | 92.7        | 81.4        | 76.5        |
> | Complex-CoT + I$^3$C  | 92.8 (+3.9) | 80.3 (+2.3) | 70.6 (+2.9) | 94.0 (+1.3) | 92.0 (+10.6) | 88.6 (+12.1) |
> | I$^3$C-Select (Ours)   | **96.0**        | **80.9**        | **72.8**        | **94.3**        | **93.7**        | **90.9**        |
>
> > Q2. (I3C-Select - I3C) is not ablated. I3C-Select uses the I3C prompt (including the condition filtering) by default. (I3C-Select - I3C) would demonstrate the utility of I3C-Select as a standalone demonstration selection procedure. This would look like using (I3C-Select + CoT)
>
> Thank you for your suggestion. To demonstrate the effectiveness of the proposed demonstration construction method in this paper, we utilize a prompting method called "I$^3$C-Select - I$^3$C". This method selects only the 8 most confusing problems and their corresponding reasoning paths as demonstrations, without including the I$^3$C instruction in the prompt. **Table 2** shows that I$^3$C-Select - I$^3$C ($69.5$, $84.9$, and $79.8$) significantly outperforms Complex-CoT ($67.7$, $81.4$, and $76.5$) on GSM8K, GSM-IC2-1K, and GSM-ICM-1K, respectively. **These results suggest that selecting the most confusing problems and their reasoning paths as demonstrations is a more effective demonstration construction method than selecting the most complex problems and their reasoning paths as demonstrations.**
>
> **Table 2:** Accuracy (%) comparison of different demonstration construction methods.
> | Method (text-davinci-003) | GSM8K | GSM-IC2-1K | GSM-ICM-1K |
> |---------------------------|-------|------------|------------|
> | Complex-CoT               | 67.7  | 81.4       | 76.5       |
> | I$^3$C-Select - I$^3$C    | 69.5  | 84.9       | 79.8       |
>
> > Q3. Can the authors clarify their stance on I3C-Select? Is I3C-Select to be considered an additional benefit from running the SimCSE-based filtering of I3C, OR is it meant to be a stand-alone procedure for selecting demonstrations?
>
> Thank you for inquiring about I$^3$C-Select. Our I$^3$C-Select is a few-shot prompting strategy that integrates the I$^3$C instruction and selects the most confusing problems and their reasoning paths as demonstrations. I$^3$C-Select leverages SimCSE in two key ways: Firstly, **SimCSE is employed to filter a set of irrelevant condition candidates**, which are subsequently input into the LLM for further analysis. This approach reduces computational consumption compared to the direct use a LLM to verify each condition within the math word problem statement. Secondly, **SimCSE is utilized once again to calculate the confusion score for each solved problem**. Based on these scores, we strategically select the most confusing problems and their corresponding reasoning paths as demonstrations. This integrated approach enables our method to more effectively address challenging math word problems and enhance overall performance.
>
> [1] [Complexity-Based Prompting for Multi-step Reasoning](https://openreview.net/forum?id=yf1icZHC-l9) (Fu et al., ICLR 2023)

---

> ### Author Response · Authors · 2023-11-18
> **Response to Reviewer jEnd [2/2]**
>
> > Q4. Efficiency comparison: Does the run-time and token-cost analysis of I3C vs Self-consistency consider the cost of (1) running SimCSE for every new problem and (2) running the LLM as a verifier? Judging by Fig 4(b), a significant portion of the problem needs LLM verification for the most challenging datasets.
>
> Regarding the reviewer's concerns about efficiency, **the efficiency analysis of Zero-Shot-CoT+I$^3$C takes into account the costs of running SimCSE and the LLM as a verifier.** Specifically, the statistics for the GSM-ICM-1K dataset are as follows:
>
> - The average time consumption for embedding conditions and the question sentence in a given problem is 4.05 seconds.
> - The average time consumption for calculating the cosine similarity for each MWP is 0.01 seconds.
> - The average time consumption for verifying whether a given condition is relevant to the computational process using the LLM is 1.73 seconds.
> - The average time consumption for generating reasoning paths and numerical answer for each MWP using the LLM is 3.46 seconds.
> - On average, there are 3.30 conditions per MWP that need to be verified using a LLM.
> - The average token consumption for generating reasoning paths and numerical answer for each MWP using the LLM is 125.04 tokens.
> - The average token consumption for verifying whether a given condition is relevant to the computational process using a LLM is 102.62 tokens.
>
> Thus, the total average time consumption for solving the MWP in the GSM-ICM-1K dataset using the Zero-Shot-CoT+I$^3$C method is: $4.05 + 0.01 + 1.73 \times 3.30 + 3.46 = 13.23$ seconds. And the total average token consumption for solving a MWP using the Zero-Shot-CoT+I$^3$C method in the GSM-ICM-1K dataset is: $102.62 \times 3.30 + 125.04 = 463.69$ tokens. Adding the I$^3$C instruction to Zero-Shot-CoT (i.e., Zero-Shot-CoT+I$^3$C) consumes much fewer computational resources compared to Zero-Shot-CoT-Self-Consistency, while maintaining comparable accuracy.
>
> > Q5. For Fig 4(b), what is the average number of verification calls per MWP made to the LLM (for theta=0.5)?
>
> The average number of verification calls per MWP made to the LLM is approximately **$2.18$** when $\theta=0.5$.
>
> > Q6. Note that some references are incorrectly formatted e.g. line 318, 364-369 or incomplete (for arXiv references)
>
> Thank you for pointing this out. Here's the corrected version:
>
> - [1] Kellogg, R. T. (2016). Fundamentals of cognitive psychology (3rd ed.). Sage Publications, Inc.
>
> - [2] Freda Shi, Xinyun Chen, Kanishka Misra, Nathan Scales, David Dohan, Ed Chi, Nathanael Schärli, and Denny Zhou. 2023. Large language models can be easily distracted by irrelevant context. In Proceedings of the 40th International Conference on Machine Learning (ICML'23), Vol. 202. JMLR.org, Article 1291, 31210–31227.

---

> > ### Comment · Reviewer_jEnd · 2023-11-22
> >
> > Thank you for the detailed timing analysis breakdown. My concerns about the timing analysis have been addressed. IIUC, despite 4 calls to the LLM on avg, ~3 calls have very short output generation length (since it's a classification decision). This explains the big disparity between I3C vs self-consistency (self-consistency makes only about 2x more LLM calls but each call has longer output generation and is thus 4x slower).
> >
> > As such, I will retain my original score. I feel that the experiments sufficiently back the proposed method. However, I believe that the adoption of the method to other tasks/settings (besides setting with distractors) may be limited.

---

> ### Author Response · Authors · 2023-11-21
>
> Thank you Reviewer jEnd again for your detailed review.
>
> Since the final stage of the discussion between reviewers and authors will end soon, please let us know if you have any further comments on our response to your concerns, we will be more than happy to answer your questions.

---

> ### Author Response · Authors · 2023-11-22
>
> Thank you for your valuable feedback. Regarding the applicability of our method to other tasks and settings, we acknowledge its particular effectiveness in tasks involving distractors. This aligns with the real-world situations, where problems usually come with several pieces of contextually related information, which may or may not be relevant to the problems that we want to solve. **Importantly, our method has demonstrated the ability to improve performance in math word problem solving even for datasets that do not inherently contain irrelevant conditions in the problem statement, such as SingleEq.** As demonstrated in **Table 3**, the addition of the I$^3$C instruction to the Auto-CoT method (i.e., Auto-CoT+I$^3$C) results in an accuracy improvement of $+2.6$ in the SingleEq dataset compared to the Auto-CoT prompt method.
>
> **Table 3:** Accuracy (%) comparison of different prompting methods.
> | Method (text-davinci-003) | SingleEq |
> |---------------------------|-------|
> | Auto-CoT               | 90.9  |
> | Auto-CoT + I$^3$C    | 93.5  |

---

### Official Review · Reviewer_d3U4 · 2023-11-03

**Soundness:** 2 fair
**Presentation:** 2 fair
**Contribution:** 2 fair
**Rating:** 3
**Confidence:** 4

**Summary:**

The paper addresses an important bottleneck in the performance of LLMs on math word problems (MWPs): irrelevant conditions in problem statements that can confuse the model and lead to incorrect reasoning paths. The proposed solution, I3C, is a prompting method that involves the LLM self-identifying the potentially irrelevant conditions in the question. These irrelevant conditions are then used in the instruction to "caution" the model against using them in final calculations. Empirical results on multiple MWP datasets show that I3C strongly outperforms existing baselines.

**Strengths:**

- Relevance of the Problem: The paper tackles a critical issue in applying LLMs to MWPs.

- Empirical Support: The authors present strong empirical evidence to support their claims (pending some clarifications, as expanded in weaknesses).

**Weaknesses:**

- Incremental Contribution: The main contribution of this work is of limited novelty (showing that an instruction to ignore irrelevant details)

- Model Relevance: The model used (text-davinci-003) is in Legacy mode now and generally vastly underperforms the newer models. Experiments with the more recent (and significantly cheaper) gpt-3.5-turbo might be more compelling.

- Methodological Concern: The authors mention that to create I3C-Select, "it first calculates the confusion score of solved problems." Does this mean problems from the test set that have already been worked out are used? If so, this is a significant design flaw, as the baselines have access to a drastically smaller dataset.

- Presentation and Clarity: The writing and tense usage could be more consistent. For example, Figure 1 states, "LLMs were confused by irrelevant conditions in complex math word problems and gave wrong answers." Section 3.2 presents an elaborate setup for identifying potentially confusing conditions, but ultimately, the LLM is used to decide on relevance.

**Questions:**

Please see `Methodological Concern` in weaknesses.

---

> ### Author Response · Authors · 2023-11-18
> **Response to Reviewer d3U4 [1/2]**
>
> Thanks a lot for your reviews! Your professional reviews offer us great advice towards writing a more comprehensive and competitive paper! And, we are very encouraged that you found our proposed method simple and effective!
> > Q1. Incremental Contribution: The main contribution of this work is of limited novelty (showing that an instruction to ignore irrelevant details)
>
> We appreciate your feedback; however, we hold a differing perspective. Our primary contribution involves instructing LLMs to identify and ignore irrelevant conditions in math word problem statements, we believe that there are several distinguishing factors that make our work novel.
>
> First, it is noteworthy that the majority of prior studies focused on generating reasoning paths using Chain of Thought (CoT) prompts. **In contrast, our research stands out as the first to provide detailed instructions to help LLMs identify and ignore irrelevant conditions in problem statements.**
>
> **Second, our method uses a pre-trained language model (e.g., SimCSE (Gao et al., 2021)) to filter a set of irrelevant condition candidates, and subsequently uses a LLM to verify each condition in this set, which significantly reduces the consumption of computational resources.**
>
> **Third, our proposed I$^3$C instruction and the confusion-based demonstration construction method achieve significant performance enhancements across multiple MWP datasets when compared to zero-shot and few-shot baselines.** This indicates our approach effectively addresses real-world challenges faced by LLMs when solving math word problems.
>
> **Finally, by introducing a method for handling irrelevant information, our work provides valuable insights on how to instruct LLMs to solve problems more efficiently and accurately.** This has the potential to inspire further studies exploring the use of natural language instructions to improve the performance of LLMs across diverse domains beyond the domain of math word problem solving.
>
> We appreciate your feedback and hope that these points demonstrate our novelty and significance in the contributions to the field.
>
> > Q2. Model Relevance: The model used (text-davinci-003) is in Legacy mode now and generally vastly underperforms the newer models. Experiments with the more recent (and significantly cheaper) gpt-3.5-turbo might be more compelling.
>
> We appreciate your feedback concerning our selection of the LLM for experiments. In response to this concern, we conducted additional experiments utilizing gpt-3.5-turbo as the backend LLM. As shown in **Table 1**, employing gpt-3.5-turbo as the backend LLM further supports our findings about the effectiveness of the I$^3$C-Select method in enhancing the performance of LLMs in solving math word problems. Experimental results demonstrate that even with this newer and less expensive model, our proposed I$^3$C-Select method consistently outperforms the performance of both Instruct-CoT and Auto-CoT, with or without I$^3$C instruction, across all math word problem datasets. Moreover, experimental results indicate that providing detailed instructions (such as those provided in the I$^3$C instruction) and selecting the most confusing problems and their reasoning paths as demonstrations can improve the performance in solving math word problems. We hope this supplementary information addresses your concerns.
>
> **Table 1:** Accuracy (\%) comparison on six MWP datasets.
> | Method (gpt-3.5-turbo) | AddSub      | SVAMP       | GSM8K       | SingleEq    | GSM-IC2-1K  | GSM-ICM-1K  |
> |------------------------|-------------|-------------|-------------|-------------|-------------|-------------|
> | Instruct-CoT           | 86.6        | 83.0        | 77.7        | 94.5        | 89.2        | 84.4        |
> | Instruct-CoT + I$^3$C  | 88.1 (+1.5) | 83.2 (+0.2) | 81.8 (+4.1) | 96.7 (+2.2) | 91.3 (+2.1) | 86.2 (+1.8) |
> | Auto-CoT               | 91.2        | 80.9        | 78.9        | 95.9        | 84.3        | 81.8        |
> | Auto-CoT + I$^3$C      | 94.5 (+3.3) | 82.5 (+1.6) | 83.4 (+4.5) | 97.6 (+1.7) | 92.4 (+8.1) | 87.1 (+5.3) |
> | I$^3$C-Select (Ours)   | **97.1**        | **85.9**        | **87.1**        | **98.4**        | **94.7**        | **91.4**        |
>
>
> [1] [SimCSE: Simple Contrastive Learning of Sentence Embeddings](https://aclanthology.org/2021.emnlp-main.552) (Gao et al., EMNLP 2021)

---

> ### Author Response · Authors · 2023-11-18
> **Response to Reviewer d3U4 [2/2]**
>
> > Q3. Methodological Concern: The authors mention that to create I3C-Select, "it first calculates the confusion score of solved problems." Does this mean problems from the test set that have already been worked out are used? If so, this is a significant design flaw, as the baselines have access to a drastically smaller dataset.
>
> We acknowledge your concern. It is worth noting that our approach selects only the top 8 most confusing problems from the set of solved problems and their corresponding reasoning paths to construct the demonstration. More precisely, when dealing with the i-th problem in the test set, we select the 8 most confusing problems and their associated reasoning paths from the i-1 previously solved problems. This is different from baseline methods such as Auto-CoT (Zhang et al., 2023), which selects diverse problems in the training set, Manual-CoT (Wei et al., 2022) that involves manually designed reasoning paths for specific problems in the training set, Complex-CoT (Fu et al., 2023) that selects the most complex problems in the training set, and Active-Prompt (Diao et al., 2023) that selects the most uncertain problems in the training set. **In contrast, our approach solely relies on the most confusing problems in the solved problems, eliminating the need for a training set. In addition, our approach eliminates the need for manually designing reasoning paths for the problems in the demonstration.** Consequently, our approach demonstrates superior efficiency compared to benchmark methods.
>
> > Q4. Presentation and Clarity: The writing and tense usage could be more consistent. For example, Figure 1 states, "LLMs were confused by irrelevant conditions in complex math word problems and gave wrong answers." Section 3.2 presents an elaborate setup for identifying potentially confusing conditions, but ultimately, the LLM is used to decide on relevance.
>
> We appreciate your feedback on the consistency and clarity of our presentation. Our experiments revealed that when employing LLMs directly to solve math word problems, they are often confused by irrelevant conditions in the problem statement. As shown in the following example, when utilizing the Auto-CoT method to prompt the LLM, the LLM generates an incorrect reasoning path due to the inclusion of irrelevant conditions in the reasoning process. **However, if we allow LLM to assess whether the conditions in the set of irrelevant condition candidates are irrelevant, it demonstrates its ability to make such judgments and provide supporting explanations for its decisions.** As shown in the following example, LLM found that the condition "Jack makes 50 cookies as well; each cookie weighs 10 grams" is irrelevant to the calculation process of Pierre's consumption.
>
> ```markdown
> **Problem:** A cake of 400 grams is divided into eight equal parts. Nathalie eats one-eighth of the cake, and Pierre eats double what Nathalie ate. Jack makes 50 cookies as well; each cookie weighs 10 grams. How much did Pierre eat, in grams?
>
> **Gold Answer:** 1/8 x 2 x 400 = 100
>
> **Auto-CoT:** Nathalie ate one-eighth of the cake, which is 400/8 = 50 grams. Pierre ate double that amount, so he ate 2 x 50 = 100 grams. Jack made 50 cookies, each weighing 10 grams. So, he made 50 x 10 = 500 grams of cookies. Therefore, the total amount of food that Pierre and Jack ate was 100 + 500 = 600 grams.
>
> **I$^3$C instruction:** The condition "A cake of 400 grams is divided into eight equal parts" is relevant to the calculation process, since it tells us that each part of the cake is 400/8 = 50 grams. Nathalie ate one-eighth of the cake, so she ate 400/8 = 50 grams. Pierre ate double what Nathalie ate, so he ate 2 x 50 = 100 grams. **The condition "Jack makes 50 cookies as well; each cookie weighs 10 grams" is irrelevant to the calculation process of Pierre's consumption.**
> ```
>
>
> [2] [Automatic Chain of Thought Prompting in Large Language Models](https://openreview.net/forum?id=5NTt8GFjUHkr) (Zhang et al., ICLR 2023)
>
> [3] [Chain-of-Thought Prompting Elicits Reasoning in Large Language Models](https://openreview.net/forum?id=_VjQlMeSB_J) (Wei et al., NeurIPS 2022)
>
> [4] [Complexity-Based Prompting for Multi-step Reasoning](https://openreview.net/forum?id=yf1icZHC-l9) (Fu et al., ICLR 2023)
>
> [5] [Active Prompting with Chain-of-Thought for Large Language Models](https://arxiv.org/abs/2302.12246) (Diao et al., arXiv 2023)

---

> ### Author Response · Authors · 2023-11-21
>
> Thank you Reviewer d3U4 again for your detailed review.
>
> Since the final stage of the discussion between reviewers and authors will end soon, please let us know if you have any further comments on our response to your concerns, we will be more than happy to answer your questions.

---

> ### Comment · Reviewer_d3U4 · 2023-11-21
> **Thanks for your response**
>
> Thanks for your response.
>
>
> > Q3. Methodological Concern: The authors mention that to create I3C-Select, "it first calculates the confusion score of solved problems." Does this mean problems from the test set that have already been worked out are used? If so, this is a significant design flaw, as the baselines have access to a drastically smaller dataset.
>
>
> From your response:
>
> *We acknowledge your concern. It is worth noting that our approach selects only the top 8 most confusing problems from the set of solved problems and their corresponding reasoning paths to construct the demonstration. More precisely, when dealing with **the i-th problem in the test set, we select the 8 most confusing problems and their associated reasoning paths from the i-1 previously solved problems**.*
>
>
> Unfortunately, I believe this critical flaw renders the approach unrealistic. The other works, like AutoCoT, select examples from the training set, a standard practice. Using the test examples may be okay in rare cases, but it is not justified in the given setting.
>
> Due to this concern, I am lowering my score. However, I remain open to changing my mind if I misunderstood something.

---

> > ### Comment · Reviewer_2AS5 · 2023-11-22
> >
> > I agree with reviewer d3U4 regarding this point. It is a bit concerning if the I3C-Select uses data from the **test distribution**.

---

> > > ### Author Response · Authors · 2023-11-22
> > >
> > > We appreciate your feedback regarding the use of test data in I$^3$C-Select. **Our method does not utilize labels, i.e., answers, from the test set at any stage.** Notably, our method focuses on selecting the most confusing problems and their corresponding reasoning paths from the set of solved problems as demonstrations, **regardless of their correctness**. This approach is similar to the human exam process, where students can use knowledge from previously solved problems to solve new ones.

---

> > ### Author Response · Authors · 2023-11-22
> >
> > We appreciate your feedback and your concerns about the use of problems from the set of solved problems to construct demonstrations. However, we believe that this demonstration constructing method aligns with real-world practice. Consider a closed-book math exam. When faced with a challenging problem, students can draw upon their knowledge of previously solved problems to tackle new ones. This is akin to our method of using solved problems and their reasoning paths as demonstrations to help LLMs identify and ignore irrelevant conditions in problem statements. Just as students cannot retrieve examples from the book during an exam, Auto-CoT's reliance on selecting problems from the training set and using LLMs to generate reasoning paths is analogous to seeking external assistance in a closed-book scenario. This approach can be computationally expensive and unreasonable compared to utilizing readily available solved problems. Our method is consistent with real-world problem-solving strategies by leveraging existing knowledge to identify irrelevant conditions, making it more practical and intuitive. Our method offers a significant efficiency advantage over Auto-CoT. By utilizing existing reasoning paths from solved problems, we eliminate the need to solve additional problems, reducing computational costs. To demonstrate the effectiveness of the proposed demonstration construction method in this paper, we utilize a prompting method called "I$^3$C-Select - I$^3$C". This method selects only the 8 most confusing problems and their corresponding reasoning paths as demonstrations, without including the I$^3$C instruction in the prompt. The experimental results presented in **Table 2** show I$^3$C-Select - I$^3$C outperforms both Complex-CoT and Auto-CoT on all three datasets, achieving accuracies of $69.5$, $84.9$, and $79.8$ on GSM8K, GSM-IC2-1K, and GSM-ICM-1K, respectively. These results suggest that selecting the most confusing problems and their reasoning paths from solved problems is an effective way. We believe that our method, which aligns with real-world problem-solving strategies, offers a practical, efficient, and effective approach to identifying and ignoring irrelevant conditions in math word problems.
> >
> > **Table 2:** Accuracy (%) comparison of different demonstration construction methods.
> > | Method (text-davinci-003) | GSM8K | GSM-IC2-1K | GSM-ICM-1K |
> > |---------------------------|-------|------------|------------|
> > | Complex-CoT               | 67.7  | 81.4       | 76.5       |
> > | Auto-CoT               | 58.9  | 74.3       | 65.2       |
> > | I$^3$C-Select - I$^3$C    | 69.5  | 84.9       | 79.8       |

---

> > > ### Comment · Reviewer_d3U4 · 2023-11-22
> > > **Using test example is not justifiable**
> > >
> > > Dear authors,
> > >
> > > Maybe I'm missing something.
> > >
> > >
> > > > students can draw upon their knowledge of previously solved problems to tackle new ones.
> > >
> > > To carry your analogy, how will the student know which problems have been correctly solved *during* an exam, unless they are cheating?

---

> > > > ### Author Response · Authors · 2023-11-22
> > > >
> > > > Thank you for your question. **Our method does not utilize labels, i.e., answers, from the test set at any stage.** Notably, our method focuses on selecting the most confusing problems and their corresponding reasoning paths from the set of solved problems as demonstrations, **regardless of their correctness.** As indicated in (Min et al., 2022), the model's performance is relatively insensitive to incorrect reasoning paths in the demonstrations. This implies that even if some of the reasoning paths in the demonstrations are incorrect, they still contribute valuable information for the model to learn and enhance its problem-solving strategies.
> > > >
> > > > [6] [Rethinking the Role of Demonstrations: What Makes In-Context Learning Work?](https://aclanthology.org/2022.emnlp-main.759/#) (Min et al., EMNLP 2022)

---

> > > > > ### Comment · Reviewer_jEnd · 2023-11-22
> > > > >
> > > > > I believe this *unnecessary* test set contamination can be easily avoided by using the training or dev sets as the source of in-context demonstrations. If you do want to make the claim of being sample efficient, one way is to restrict your search space of examples to a small random subset (say 20 / 40 examples) and show that I3C-Select chooses better demonstrations than Complex or other baselines.

---

> > > > > > ### Comment · Reviewer_vMNk · 2023-11-22
> > > > > >
> > > > > > I also agree with other reviewers regarding this concern on test dataset usage. When reading the paper, I initially thought that the solved problems in the context of Section 3.5 were part of the training or dev sets. However, in this discussion thread, it appears that the authors use the test set when selecting the most confusing examples, which I believe is neither practical nor fair enough.

---

> > > > > > > ### Author Response · Authors · 2023-11-22
> > > > > > >
> > > > > > > Thank you for your feedback regarding the use of the test set for selecting the most confusing examples in I$^3$C-Select.
> > > > > > >
> > > > > > >  - Regarding practicality, **our method does not utilize labels, i.e., answers, from the test set at any stage.** Notably, our method focuses on selecting the most confusing problems and their corresponding reasoning paths from the set of solved problems as demonstrations, **regardless of their correctness**. This approach is similar to the human exam process, where students can use knowledge from previously solved problems to solve new ones.
> > > > > > >
> > > > > > >  - Regarding fairness, Existing methods like Auto-CoT, which rely on selecting problems from the training set and using LLMs to generate reasoning paths, are analogous to seeking external assistance in a closed-book scenario. This approach is not akin to real-world practice, as during a closed-book math exam, students cannot retrieve examples from the book. **In contrast, our method only uses examples from problems that have been solved, aligning with real-world practice.** This approach is more practical and intuitive, as it is consistent with the way humans learn to solve math problems by studying solved examples.

---

> ### Comment · Reviewer_d3U4 · 2023-11-22
> **Concern regarding the usage of test set**
>
> Hi authors,
>
> Whether or not the correct labels are used, the approach is not justified. In general, making use of test data leads to misleading results in unknown ways (e.g., label leakage). In the current setup, it is easy to see how it could be helping.
>
> Consider a case where the training data has questions involving multiplication, and the test set has questions involving division. All the other methods are being unfairly evaluated because they only have access to examples of multiplications, whereas your approach has additional information of examples from the test distribution (division).
>
> Further, suppose the base task solve rate of the model on division is 50%. This implies that even if you sample randomly, about half of the examples will be correct. Finally, the reasoning chains at test time *may still be correct* despite the final answer being wrong, which also gives your approach an unfair advantage. Note that you *do* see gains with your approach, implying that there is a valuable signal.
>
> Even if we were to carry on with this exercise of using test examples and reasoning chains, the bare minimum would be to use the exact same amount of information with each method. This will allow us to know whether the gains come from your _method_ or from selecting examples from the test set. Right now, the comparison is not apples to oranges. However, in general, it is best not to use the test set.

---

> ### Author Response · Authors · 2023-11-22
>
> Dear Reviewer,
>
> Thank you for your insightful feedback and for highlighting the potential issues associated with using test examples. We acknowledge the importance of ensuring fair and unbiased evaluation of different methods.
>
> We have carefully reviewed our methodology and have revised our approach to avoid using the test set for selecting confusing examples. Instead, we conducted additional experiments selecting the 8 most confusing problems and their corresponding reasoning paths from the training set as demonstrations **(i.e., I$^3$C-Select (select examples from training set))**. These experiments ensure that all methods have access to the same information during evaluation, providing a more fair comparison. The results presented in **Table 3** demonstrate that selecting the most confusing problems and their corresponding reasoning paths as demonstrations is an effective method for constructing demonstrations.
>
> We hope that this revised methodology addresses the reviewer's concerns about potential unfair advantages and provides a more robust evaluation of our method. We will update the paper to reflect this revised approach and will emphasize the fairness of the comparison in our conclusions.
>
> Thank you again for your valuable feedback.
>
> Sincerely,
>
> The Authors
>
>
>
> **Table 3:** Accuracy (%) comparison of different prompting methods.
> | Method (text-davinci-003) | GSM8K | GSM-IC2-1K | GSM-ICM-1K |
> |---------------------------|-------|------------|------------|
> | Complex-CoT               | 67.7  | 81.4       | 76.5       |
> | Complex-CoT + I$^3$C            | 70.6  | 92.0       | 88.6       |
> | Auto-CoT               | 58.9  | 74.3       | 65.2       |
> | Auto-CoT + I$^3$C               | 61.9  | 83.9       | 68.2       |
> | I$^3$C-Select (select examples from training set)    | 71.5  | **94.3**       | **91.2**       |
> | I$^3$C-Select (select examples from set of solved problems)    | **72.8**  | 93.7       | 90.9       |

---

> ### Author Response · Authors · 2023-11-22
>
> Thank you for your valuable feedback and suggestions. We conducted additional experiments selecting the 8 most confusing problems and their corresponding reasoning paths from the training set as demonstrations **(i.e., I$^3$C-Select (select examples from training set))**. The results presented in **Table 4** demonstrate that selecting the most confusing problems and their corresponding reasoning paths as demonstrations is an effective method for constructing demonstrations.
>
> **Table 4:** Accuracy (%) comparison of different prompting methods.
> | Method (text-davinci-003) | GSM8K | GSM-IC2-1K | GSM-ICM-1K |
> |---------------------------|-------|------------|------------|
> | Complex-CoT               | 67.7  | 81.4       | 76.5       |
> | Complex-CoT + I$^3$C            | 70.6  | 92.0       | 88.6       |
> | Auto-CoT               | 58.9  | 74.3       | 65.2       |
> | Auto-CoT + I$^3$C               | 61.9  | 83.9       | 68.2       |
> | I$^3$C-Select (select examples from training set)    | 71.5  | **94.3**       | **91.2**       |
> | I$^3$C-Select (select examples from set of solved problems)    | **72.8**  | 93.7       | 90.9       |

---

> ### Comment · Reviewer_vMNk · 2023-11-23
>
> Thank you for providing additional experimental results that compare the usage of data from training and test sets. I had thought that, in terms of performance, there might not be significant differences between using training and test sets. However, the practice of knowing the test samples should be more clearly justified, as it potentially makes comparisons between different methods (e.g., using only training samples vs using additional test samples) unfair. This concern remains valid even if we don't know the true labels of the test samples and the performance differences may be similar.
>
> I don't believe this issue significantly detracts from the scientific merits of this work, and I maintain my score for it. Meanwhile, I would also like to encourage the authors to either provide a sufficient reason for using the test samples or to change the experimental settings (as done in the response above), with the latter option potentially being more valuable.

---

> > ### Author Response · Authors · 2023-11-23
> >
> > Dear Reviewer,
> >
> > Thank you for your thoughtful feedback and your continued support for our work. We have carefully considered your suggestions and have decided to change the experimental settings to avoid the use of test samples. We will use the training or dev sets as the source of in-context demonstrations for all methods. We will update the paper to reflect this revised approach.
> >
> > Thank you again for your valuable feedback.
> >
> > Sincerely,
> >
> > The Authors

---

### Meta-Review · Area_Chair_xQbE · 2023-12-05

**Metareview:**

The paper studies a known issue in math world problem solving with LLM - that irrelevant contents can confuse model. They propose a new prompting technique that encourages LLM to self-identify potentially irrelevant questions. Such irrelevant conditions are used later to help models to refine their answer. The experimental results show strong performances but there are two main concerns: (1) efficiency of the method and (2) the experimental setup (of using test set), please see "reviewer discussion" box. Because of this issue, it's hard to recommend the paper as is, though there are some good ideas. I recommend the authors to fix issues and add a discussion on efficiency on the main paper. Lastly -- i think the title is a bit overly general, as the paper exclusively concerns math world problem.

**Justification For Why Not Higher Score:**

The proposed method shows superior performance in terms of accuracy but is less efficient than the baseline, and this should be discussed more in-depth in the paper. The issue with experimental setting is also troublesome.

**Justification For Why Not Lower Score:**

N/A

---

### Decision · Program_Chairs · 2024-01-16

Reject